# The linear ubiquitin chain assembly complex regulates TRAIL-induced gene activation and cell death

Elodie Lafont[1], Chahrazade Kantari-Mimoun[1,†], Peter Draber[1,†], Diego De Miguel[1], Torsten Hartwig[1], Matthias Reichert[1], Sebastian Kupka[1], Yutaka Shimizu[1], Lucia Taraborrelli[1], Maureen Spit[1], Martin R Sprick[2] & Henning Walczak[1,*] iD

## Abstract

The linear ubiquitin chain assembly complex (LUBAC) is the only known E3 ubiquitin ligase which catalyses the generation of linear ubiquitin linkages *de novo*. LUBAC is a crucial component of various immune receptor signalling pathways. Here, we show that LUBAC forms part of the TRAIL-R-associated complex I as well as of the cytoplasmic TRAIL-induced complex II. In both of these complexes, HOIP limits caspase-8 activity and, consequently, apoptosis whilst being itself cleaved in a caspase-8-dependent manner. Yet, by limiting the formation of a RIPK1/RIPK3/MLKL-containing complex, LUBAC also restricts TRAIL-induced necroptosis. We identify RIPK1 and caspase-8 as linearly ubiquitinated targets of LUBAC following TRAIL stimulation. Contrary to its role in preventing TRAIL-induced RIPK1-independent apoptosis, HOIP presence, but not its activity, is required for preventing necroptosis. By promoting recruitment of the IKK complex to complex I, LUBAC also promotes TRAIL-induced activation of NF-κB and, consequently, the production of cytokines, downstream of FADD, caspase-8 and cIAP1/2. Hence, LUBAC controls the TRAIL signalling outcome from complex I and II, two platforms which both trigger cell death and gene activation.

**Keywords** cell death; LUBAC; NF-κB; TRAIL; ubiquitin
**Subject Categories** Autophagy & Cell Death; Post-translational Modifications, Proteolysis & Proteomics
**The EMBO Journal (2017) 36: 1147–1166**

See also: **H Wajant** (May 2017)

## Introduction

The tumour necrosis factor (TNF)-related apoptosis-inducing ligand (TRAIL) is a member of the TNF superfamily of cytokines identified based on its homology to CD95 ligand (CD95L) and TNF (Wiley *et al*, 1995; Pitti *et al*, 1996). Systemic treatment of tumour-bearing mice with recombinant TRAIL can efficiently kill tumour cells without inducing any toxicity and was suggested to be a promising new avenue for cancer treatment (Ashkenazi *et al*, 1999; Walczak *et al*, 1999). However, the therapeutic benefit of TRAIL receptor (TRAIL-R) agonists in clinical studies has been limited as cancer cells develop resistance and, in some instances, TRAIL promotes tumorigenesis (Micheau *et al*, 2013; von Karstedt *et al*, 2015). This points out the need for designing novel TRAIL-R agonists (Tuthill *et al*, 2015; de Miguel *et al*, 2016) and improving the understanding of TRAIL-induced apoptotic and non-apoptotic signalling (Lemke *et al*, 2014; Ashkenazi, 2015). Binding of TRAIL to TRAIL-R1 (also known as death receptor 4, DR4) and TRAIL-R2 (also known as DR5 or KILLER) induces formation of the death-inducing signalling complex (DISC), also termed complex I, composed of FADD, caspase-8/10 and cFLIP$_{L/S}$ (Kischkel *et al*, 1995, 2000, 2001; Sprick *et al*, 2000, 2002). Formation of the DISC/complex I leads to the activation of initiator caspases 8 and 10 which results in cleavage and activation of effector caspases 3 and 7. This in turn triggers downstream effector mechanisms of apoptosis. TRAIL-induced apoptotic signalling is tightly regulated at multiple stages. For instance, cFLIP$_S$ abrogates initiator caspase-8 activation, whereas cFLIP$_L$ enables DISC-restricted activation of caspase-8 (Hughes *et al*, 2016). Moreover, Bcl-2 family members as well as the caspase-3 and caspase-9 inhibitor XIAP modulate TRAIL-induced death (Lemke *et al*, 2014).

Besides promoting apoptosis, TRAIL has also been shown to be capable of triggering necroptosis, a caspase-independent type of cell death requiring RIPK1, RIPK3 and MLKL (Holler *et al*, 2000; Jouan-Lanhouet *et al*, 2012; Goodall *et al*, 2016). TRAIL also triggers death domain (DD)-dependent gene-activatory signalling which initiates

1   Centre for Cell Death, Cancer and Inflammation (CCCI), UCL Cancer Institute, University College London, London, UK
2   Heidelberg Institute for Stem Cell Technology and Experimental Medicine (HI-STEM gGMBH), Heidelberg, Germany
    *Corresponding author. Tel: +44 207 679 6471; E-mail: h.walczak@ucl.ac.uk
    †These authors contributed equally to this work

non-cytotoxic outcomes, including cytokine production (Azijli *et al*, 2013). The initiation of gene-activatory signalling is thought to emanate from complex II, a secondary cytoplasmic signalling complex. This complex comprises RIPK1, NEMO, TRAF2, caspase-8 and FADD but is devoid of TRAIL-R1/R2 (Varfolomeev *et al*, 2005). Furthermore, whilst cFLIP$_L$ has been proposed to modulate gene-activatory signalling, it remains unresolved whether it acts as a positive or negative regulator of gene activation (Kataoka *et al*, 2000; Kataoka & Tschopp, 2004; Golks *et al*, 2006; Kavuri *et al*, 2011). In addition, we recently showed that TRAIL/TRAIL-R exerts death domain (DD)-independent, pro-tumorigenic and pro-metastatic functions in KRAS-mutated cancer cells (von Karstedt *et al*, 2015). In summary, TRAIL can induce several distinct signalling outcomes ranging from cell death, via apoptosis or necroptosis, to gene activation, cytokine production and migration. The molecular mechanisms, especially with regard to posttranslational modification(s), which provide the basis for these different outcomes, are so far not well defined.

Ubiquitination is the attachment of the C-terminal glycine of a ubiquitin to an amino group on a target protein, usually on a lysine (K) residue (Hershko & Ciechanover, 1998). Ubiquitin itself can be the target for ubiquitination, leading to the formation of different di-ubiquitin linkages. This can be repeated to result in the formation of ubiquitin chains which can directly affect the function of target proteins but also recruit specific ubiquitin binding molecules (Husnjak & Dikic, 2012; Komander & Rape, 2012). Apart from the ε-amino groups of the seven lysines on the target-attached ubiquitin, also the α-amino group of the N-terminal methionine of ubiquitin can be used to create an inter-ubiquitin linkage. In the latter case, the result is the formation of the so-called linear or methionine 1 (M1) linkage. The linear ubiquitin chain assembly complex (LUBAC), composed of SHARPIN, HOIL-1 and the catalytic component HOIP, is the only ubiquitin E3 currently known to form such chains *de novo* (Kirisako *et al*, 2006; Tokunaga *et al*, 2009). LUBAC and linear ubiquitination are now considered to be important regulators of multiple signalling pathways (Walczak *et al*, 2012; Fiil & Gyrd-Hansen, 2014; Ikeda, 2015; Sasaki & Iwai, 2015; Shimizu *et al*, 2015). In TNF signalling, LUBAC stabilises the TNF receptor 1 signalling complex (TNFR1-SC) and prevents formation of the cytoplasmic death-inducing complex (Haas *et al*, 2009; Peltzer *et al*, 2014). Aberrant TNFR1-induced cell death accounts for SHARPIN deficiency-induced multi-organ inflammation (Gerlach *et al*, 2011; Ikeda *et al*, 2011; Tokunaga *et al*, 2012; Kumari *et al*, 2014; Rickard *et al*, 2014) and HOIP deficiency-induced embryonic lethality at mid-gestation (Peltzer *et al*, 2014).

Interestingly, ubiquitination was recently identified as an additional level of regulation of TRAIL signalling as cullin-3-mediated K63-linked ubiquitination of caspase-8 was shown to promote its activation at the TRAIL DISC whilst TRAF2-dependent K48-linked ubiquitination was found to trigger its proteasomal degradation (Jin *et al*, 2009; Gonzalvez *et al*, 2012). In addition, pharmacologic depletion of the ubiquitin E3s cIAP1/2 by SMAC mimetics sensitises cells to TRAIL-induced death (Fulda *et al*, 2002; Geserick *et al*, 2009). However, whether and how LUBAC and M1-linked ubiquitination modulate TRAIL-induced signalling is currently elusive.

In this study, we identify LUBAC as a previously unrecognised component of both, the TRAIL-R-associated signalling complex I

and a secondary, cytoplasmic TRAIL-induced complex II. We demonstrate that LUBAC is crucial in regulating the balance between different outcomes of TRAIL-induced signalling, restricting cell death whilst promoting gene activation. Importantly, our study shows that complex I of TRAIL signalling does not act solely as a DISC but is also capable of initiating gene activation, thereby extending and revising the current model according to which TRAIL-induced gene-activatory signalling is limited to complex II.

# Results

## HOIP limits TRAIL-induced apoptosis and necroptosis

Since K63- and K48-linked ubiquitination regulates TRAIL-induced cell-death signalling and linear ubiquitination controls multiple immune signalling pathways, we examined whether LUBAC may be implicated in TRAIL signalling. Mouse embryonic fibroblasts (MEFs), which are poorly sensitive to TRAIL-induced cell death, were significantly sensitised in the absence of HOIP (Fig 1A and Appendix Fig S1A). We used TNF-deficient MEFs, thereby ruling out any implication of this cytokine in the observed death. Given that TRAIL can induce both apoptosis and necroptosis, we investigated the modality of TRAIL-induced death in HOIP-deficient (HOIP KO) MEFs. TRAIL-induced cell death of HOIP KO MEFs was not blocked by the caspase inhibitor z-VAD-fmk (zVAD) but by the combination of zVAD with the RIPK1 kinase inhibitor Nec-1s (Fig 1B). TRAIL-induced cleavage of caspases 8, 3 and PARP-1 and phosphorylation of MLKL were increased in HOIP KO MEFs (Fig 1C). Therefore, HOIP limits both TRAIL-induced apoptosis and necroptosis in MEFs.

We next aimed to evaluate the role of HOIP in TRAIL-induced cell death in human cancer cells. HOIP KO HeLa cells were significantly sensitised to TRAIL-induced death (Appendix Fig S1B and C). This was completely prevented by zVAD, indicating that these cells undergo apoptosis (Appendix Fig S1D). Similarly, HOIP deficiency sensitised K562 cells to TRAIL-induced apoptosis (Fig 1D and E, and Appendix Fig S1E). In accordance, TRAIL-induced cleavage of caspases 8, 10, 9 and 3 as well as of PARP-1 and Bid was enhanced in HOIP KO K562 cells (Fig 1F). In line with a previous report (Koo *et al*, 2015), RIPK3 is not expressed in K562 and HeLa cells (Appendix Fig S1F), rendering them unable to undergo necroptosis. Therefore, HOIP restricts TRAIL-induced cell death, which can be apoptotic or necroptotic depending on the cell's ability to die by these different modalities.

## LUBAC is recruited to the TRAIL-R-associated complex I in a FADD-dependent manner

We and others previously identified the TRAIL-R1/2-associated complex I as a cell-death initiating platform (Kischkel *et al*, 2000; Sprick *et al*, 2000). As HOIP prevents TRAIL-induced cell death, we hypothesised that LUBAC might form part of this TRAIL-R-associated DISC. Indeed, we found that all LUBAC components are recruited to TRAIL complex I in HeLa and A549 cells (Fig 2A and B). Reciprocally, caspase-8, TRAIL-R2 and FADD were co-immunoprecipitated with HOIP in HeLa and K562 cells upon TRAIL stimulation (Fig EV1A and B). In addition, linear ubiquitin chains

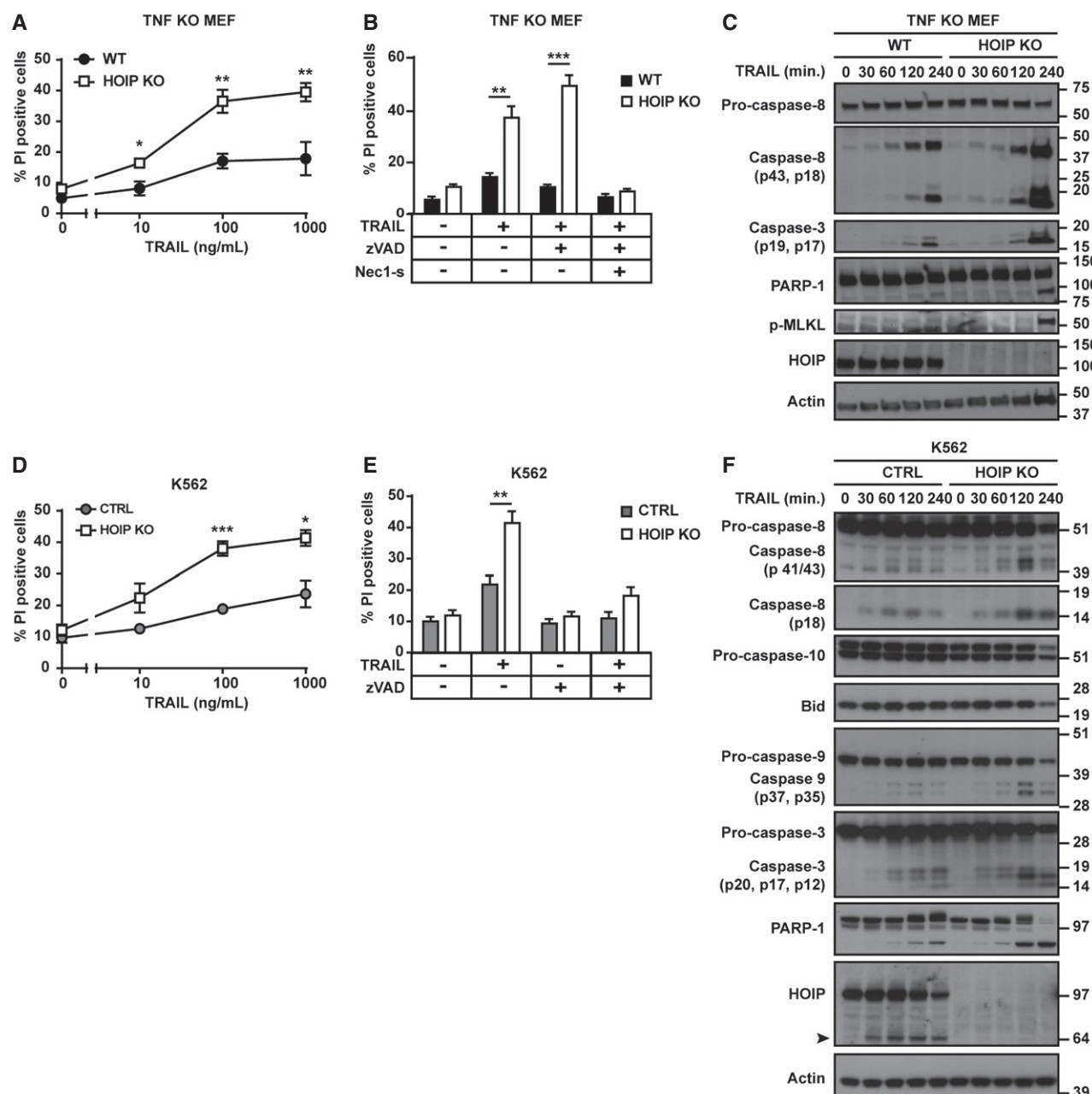

**Figure 1. HOIP limits TRAIL-induced apoptosis and necroptosis.**

A  WT and HOIP KO TNF KO MEFs were stimulated with iz-TRAIL at the indicated concentrations for 24 h (*n* = 5; mean ± SEM).
B  WT and HOIP KO TNF KO MEFs, pre-treated with zVAD and Nec-1s as indicated for 1 h, were stimulated with iz-TRAIL for 24 h (1 μg/ml) (*n* = 4; mean ± SEM).
C  WT and HOIP KO TNF KO MEFs were stimulated with iz-TRAIL (1 μg/ml) for the indicated times. Lysates were analysed by Western blot.
D  Control (CTRL) and HOIP KO K562 cells were stimulated with iz-TRAIL at the indicated concentrations for 24 h (*n* = 4; mean ± SEM).
E  Control and HOIP KO K562 cells, pre-treated with zVAD as indicated for 1 h, were stimulated with iz-TRAIL (1 μg/ml) for 24 h (*n* = 4; mean ± SEM).
F  Control and HOIP-deficient K562 cells were stimulated with iz-TRAIL (100 ng/ml) for the indicated times. Lysates were analysed by Western blot. Black arrowhead indicates the cleaved form of HOIP.

Data information: Cell death was determined after 24 h of stimulation by flow cytometry after propidium iodide (PI) labelling. *$P < 0.05$, **$P < 0.01$, ***$P < 0.001$; statistics were performed using *t*-test. See also Appendix Fig S1.

were detected in the DISC (Fig 2A and B), showing that at least one of its components is linearly ubiquitinated.

Since FADD is required for TRAIL-induced cell death (Walczak *et al*, 1997) and for formation of the TRAIL DISC (Kischkel *et al*, 2000; Sprick *et al*, 2000), we next evaluated the requirement of FADD for LUBAC recruitment. Strikingly, the TRAIL complex I in FADD KO A549 cells was devoid of LUBAC and linear ubiquitination as well as of caspase-8, RIPK1 and A20 (Fig 2C).

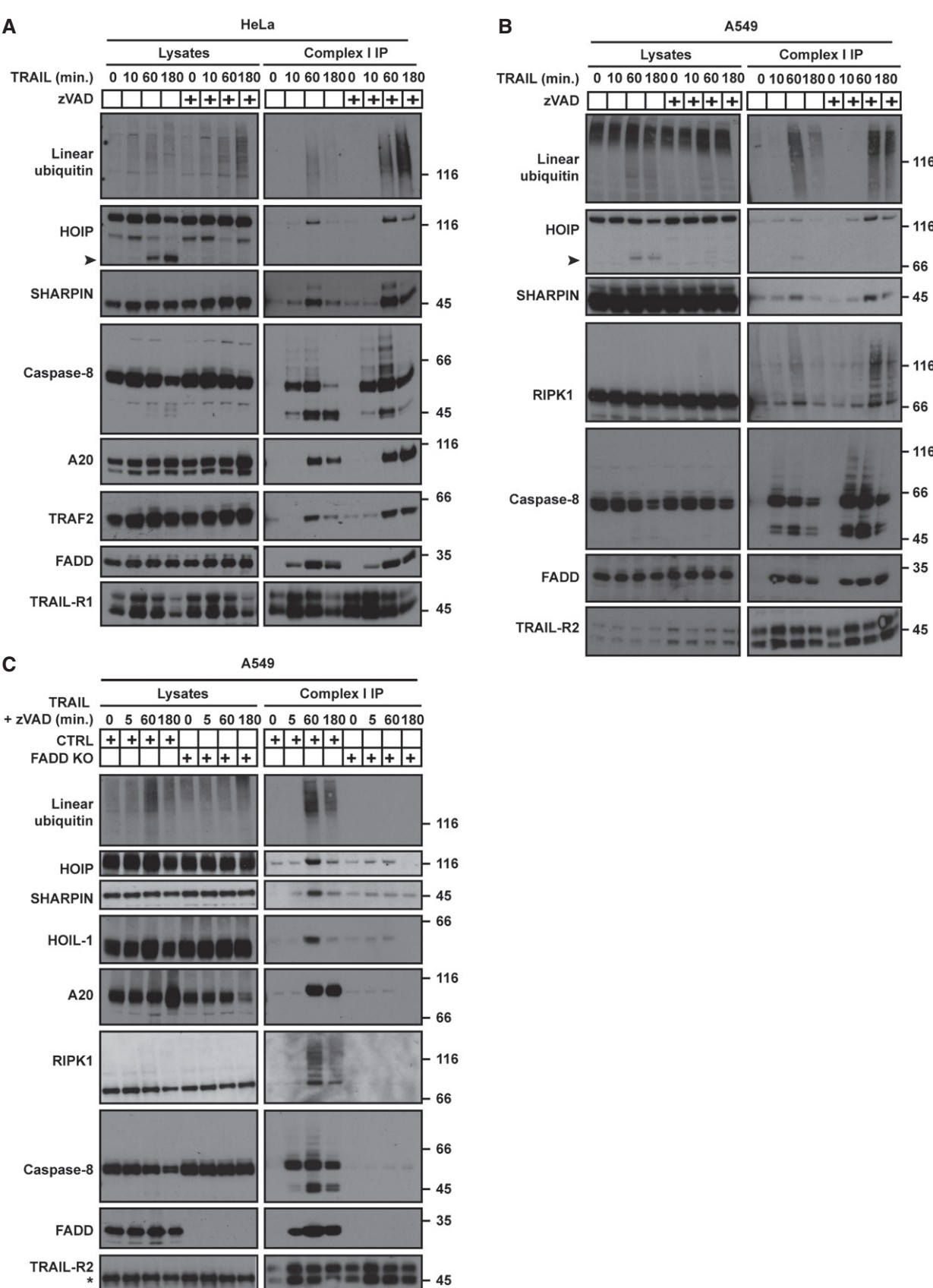

**Figure 2.**

◄

**Figure 2.  LUBAC is recruited to the TRAIL complex I in a FADD-dependent manner.**

A   HeLa cells, pre-treated for 1 h with zVAD as indicated, were treated with FLAG-lz-TRAIL (500 ng/ml) for the indicated times.
B   A549 cells, pre-treated for 1 h with zVAD as indicated, were treated with FLAG-lz-TRAIL (500 ng/ml) for the indicated times.
C   Control (CTRL) and FADD KO A549 cells, pre-treated for 1 h with zVAD, were treated with FLAG-lz-TRAIL (500 ng/ml) for the indicated times.

Data information: In all panels, the TRAIL complex I was immunoprecipitated via anti-FLAG beads and analysed by Western blot. Black arrowheads indicate the cleaved form of HOIP. * indicates an unspecific band. See also Fig EV1.

Interestingly, a shorter form of HOIP whose appearance was caspase-dependent was detected in various TRAIL-treated cell lines (Figs 2A and B, and EV1C). Thus, LUBAC is recruited to the TRAIL complex I in a FADD-dependent manner where it catalyses linear ubiquitination and where HOIP also seems to be cleaved by a caspase.

**Caspase-8-dependent cleavage of HOIP at D348/D387/D390 does not affect its apoptosis-preventing function**

We next sought to elucidate the molecular basis of HOIP cleavage upon TRAIL stimulation. We first evaluated whether active recombinant caspases cleaved HOIP *in vitro*. Strikingly, treatment of cell lysates or immuno-purified tap-tagged HOIP (HOIP-TAP) with recombinant caspases 3, 6, 8 or 10a led to the cleavage of HOIP. By contrast, the effector caspase-7, whilst being active against its endogenous substrate PARP-1, was not able to cleave HOIP (Fig 3A and Appendix Fig S2A). In MCF-7 cells, which do not express caspase-3, knock-down (KD) of caspase-8 drastically reduced HOIP cleavage induced by both, TRAIL and TNF. Conversely, KD of caspase-6 in addition to caspase-8 did not further affect HOIP cleavage. Moreover, concomitant KD of caspases 6 and 10 did not prevent HOIP cleavage (Fig 3B and Appendix Fig S2B). Therefore, contrary to caspase-6 and caspase-10, caspase-8 cleaves HOIP upon TRAIL stimulation in a cellular context.

We next determined whether caspase-3 also contributed to HOIP cleavage. Whilst KD of caspase-8 substantially reduced TRAIL- and TNF-induced HOIP cleavage, caspase-3 KD only exerted a marginal effect (Fig 3C and Appendix Fig S2C). Moreover, by comparing HCT116 WT and BAX/BAK-DKO, we observed that TRAIL-induced HOIP cleavage was similar at early time points, whereas later it was slightly reduced in HCT116 BAX/BAK-DKO in which caspase-3 was not fully activated (Fig 3D). In addition, whilst in both K562 (Fig 1F) and HT29 cells (Appendix Fig S2D) caspase-8 activation and HOIP cleavage coincided, activation of effector caspases occurred later. Interestingly, HOIP cleavage is

not limited to human cells since it also occurred in MEFs during TNF-induced apoptosis (Appendix Fig S2E). Thus, caspase-8 is the first and main caspase responsible for HOIP cleavage upon TRAIL and TNF stimulation, whilst caspase-3 marginally contributes to it at later time points.

We next sought to identify the caspase cleavage sites in HOIP to investigate the effect of HOIP cleavage on its apoptosis-preventing function. According to the apparent molecular weight of the cleavage fragment(s) observed, we narrowed down the cleavage sites within a region spanning approximately amino acids 325–400 (Fig 3A and C). Within this region, GrabCas (Backes *et al*, 2005) predicted two potential caspase-8 cleavage sites, D348 and D387. In K562 HOIP KO cells reconstituted with HOIP D348A, the higher N-terminal fragment band was still detected whilst the lower band was not. Conversely, the shorter N-terminal fragment was still visible in cells mutated for D387A (Fig 3E). Another study recently described that HOIP is cleaved at D390, in addition to D348 and D387 upon TNF stimulation (Joo *et al*, 2016). In accordance, HOIP D348A/D387A/D390A (HOIP^AAA) is completely resistant to TRAIL-induced cleavage (Fig 3E). However, reconstitution of HOIP KO K562 cells with HOIP WT or HOIP^AAA equally rescued them from TRAIL-induced death (Fig 3F).

We therefore conclude that whilst caspase-8, and to a lesser extent caspase-3, cleaves HOIP upon TRAIL and TNF stimulation, the TRAIL-induced cleavage of HOIP at D348/D387/D390 neither prevents nor enhances its apoptosis-preventing function.

**HOIP limits the activity of TRAIL-induced apoptosis- and necroptosis-mediating signalling complexes**

Since HOIP restricts TRAIL-induced apoptosis and is recruited to complex I, we hypothesised that it influences formation and/or the death-promoting activity of this complex. Indeed, activation of caspase-8 was increased in HOIP KO cells, both in complex I and in total lysates (Fig 4A). Additionally, in HOIP-deficient cells, ubiquitination of RIPK1 was reduced and its cleavage increased in complex

**Figure 3.  Caspase-8-dependent cleavage of HOIP at D348/D387/D390 does not alter its apoptosis-preventing function.** ►

A   K562 WT lysates were incubated with 3 U of the indicated recombinant active caspases for 2 h at 37°C. Samples were then analysed by Western blot with the indicated antibodies. Black arrowheads indicate the cleaved forms of HOIP.
B   MCF-7 WT cells were transfected with the indicated combinations of siRNA control, siRNA targeting caspase-8, caspase-6 or caspase-10. 72 h later, cells were treated with iz-TRAIL (1 μg/ml) for the indicated times. Lysates were analysed by Western blot. The black arrowhead indicates the cleaved form of HOIP.
C   HeLa WT cells were transfected with the indicated combinations of siRNA control, siRNA targeting caspase-8 or caspase-3. 72 h later, cells were treated with iz-TRAIL (1 μg/ml) for the indicated times. Lysates were analysed by Western blot. Black arrowheads indicate the cleaved forms of HOIP.
D   WT and BAX/BAK-DKO HCT116 cells were treated with iz-TRAIL (1 μg/ml) for the indicated times and lysates were analysed by Western blot. Black arrowhead indicates the cleaved form of HOIP.
E   K562 HOIP KO reconstituted with HOIP WT, HOIP D348A, HOIP D387A or HOIP D348A/D387A/D390A (HOIP^AAA) was incubated with iz-TRAIL (1 μg/ml) for the indicated times and lysates were analysed by Western blot. Black arrowheads indicate the cleaved forms of HOIP.
F   K562 HOIP KO reconstituted with empty vector, HOIP WT or HOIP^AAA was incubated with the indicated concentrations of iz-TRAIL for 24 h before viability was evaluated (*n* = 3; mean ± SEM). *P < 0.05, **P < 0.01, ***P < 0.001, n.s, not significant; statistics were performed using ANOVA.

Data information: See also Appendix Fig S2.

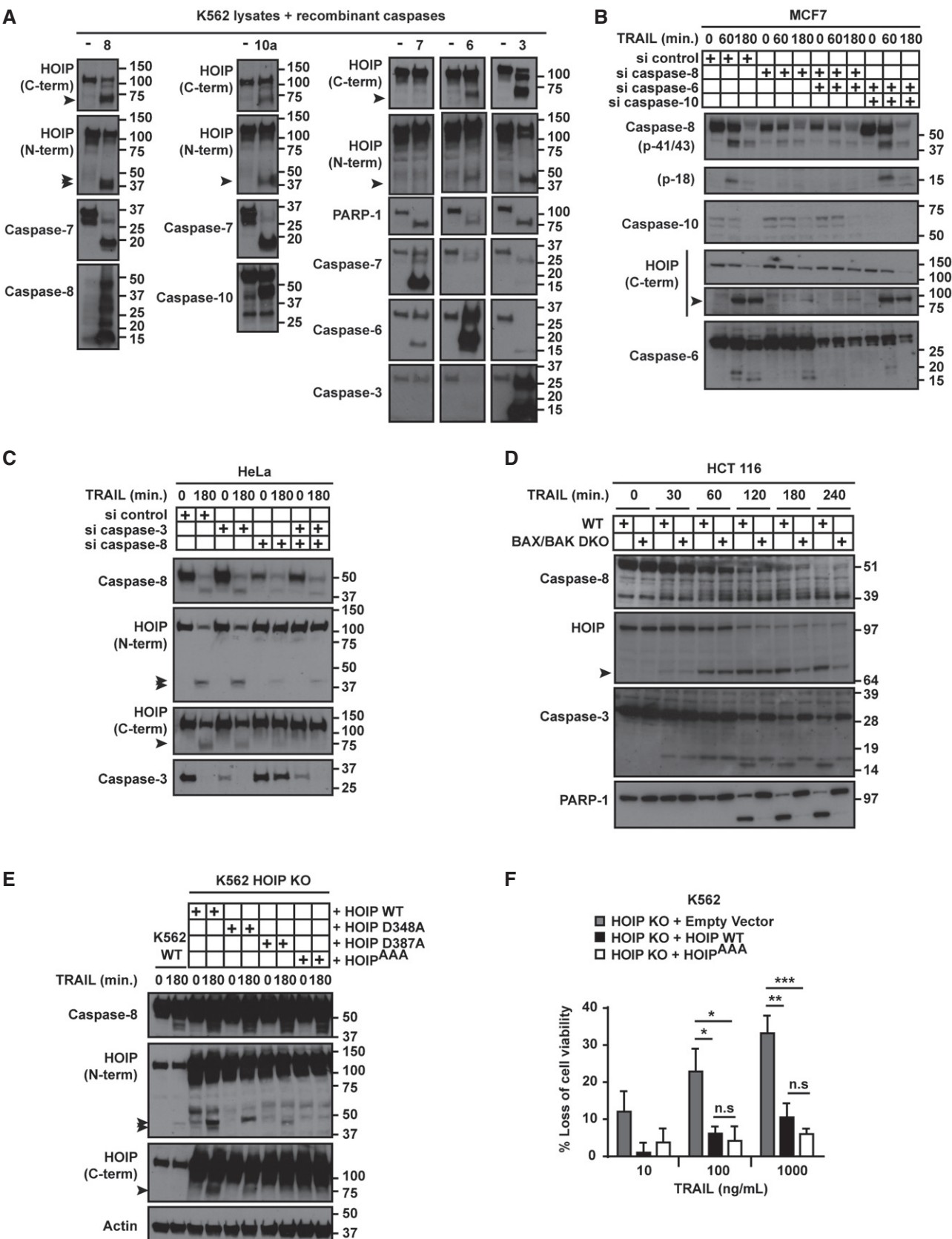

Figure 3.

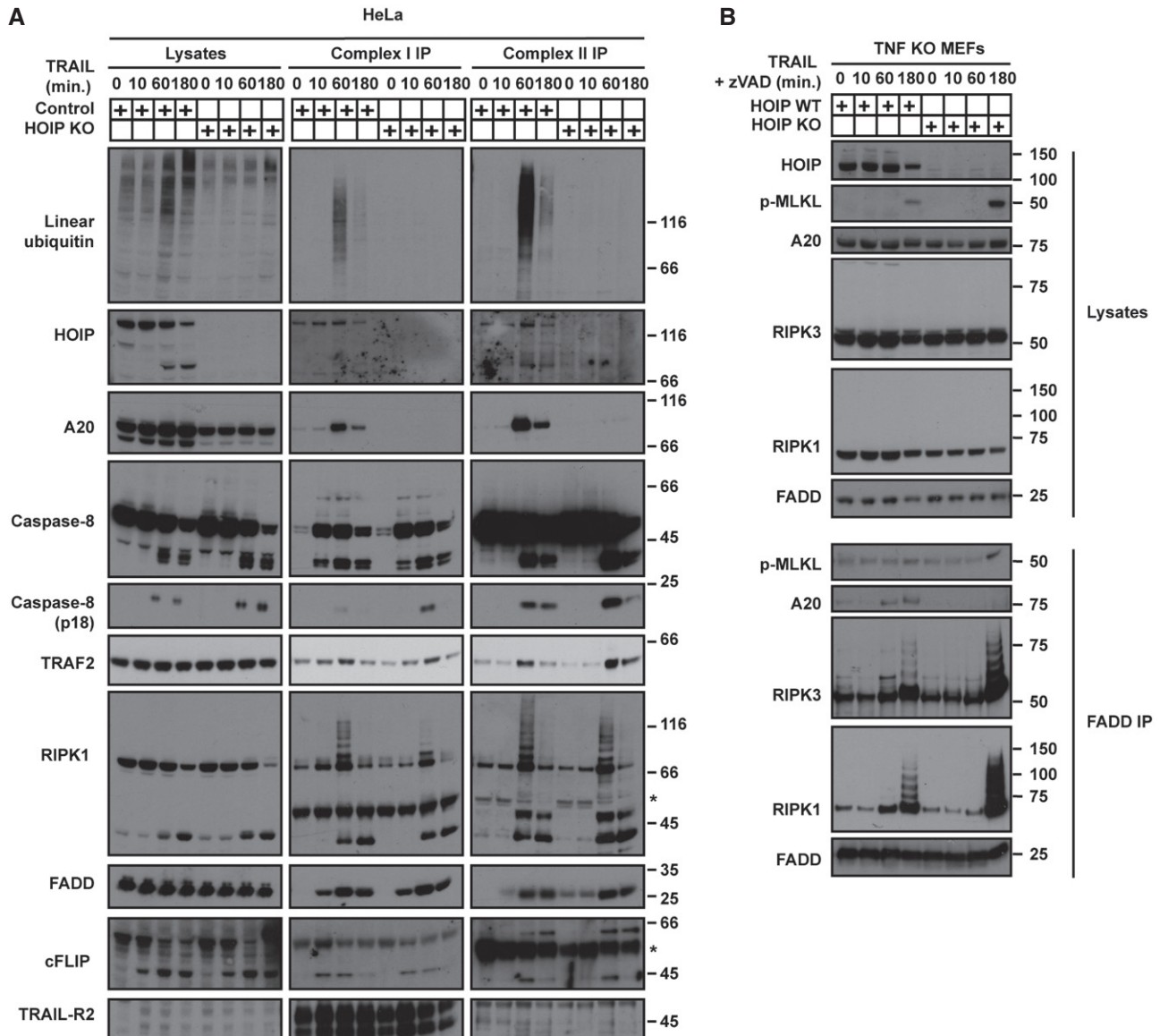

**Figure 4. HOIP limits the activity of TRAIL-induced apoptosis- and necroptosis-mediating signalling complexes.**

A   Control and HOIP KO HeLa cells were treated with FLAG-lz-TRAIL (500 ng/ml) for the indicated times. The TRAIL complex I was immunoprecipitated via anti-FLAG beads. Complex II was isolated by immunoprecipitating caspase-8 from complex I-depleted lysates. Western blot was performed using the indicated antibodies. * indicates unspecific bands.

B   WT and HOIP-deficient TNF KO MEFs, pre-treated for 1 h with zVAD, were stimulated with iz-TRAIL (1 μg/ml) for the indicated times. FADD-containing complexes were immunoprecipitated and analysed by Western blot.

Data information: See also Fig EV2.

I as well as in a secondary TRAIL-induced cytoplasmic signalling complex II, reflective of increased caspase-8 activity in both complexes (Fig 4A). Notably, RIPK1 and TRAF2 are not only components of TRAIL complex I but also of complex II (Figs 4A and EV2A).

As HOIP deficiency can also sensitise cells to necroptosis when RIPK3 and MLKL are expressed, we next studied how HOIP absence affects the necroptotic machinery. Strikingly, HOIP deficiency substantially increased formation of a FADD/RIPK1/RIPK3/MLKL-containing complex upon TRAIL/zVAD stimulation (Fig 4B). Both

RIPK1 and RIPK3 were ubiquitinated in HOIP KO cells in this complex. Moreover, MLKL phosphorylation, a modification required for its pro-necroptotic activity (Sun *et al*, 2012), was increased in HOIP KO cells (Fig 4B).

A20 is a deubiquitinase (DUB) which inhibits apoptosis (Jin *et al*, 2009; Bellail *et al*, 2012). As we and others recently found that A20 recruitment to the TNFR1-SC depends on linear ubiquitin (Tokunaga *et al*, 2012; Verhelst *et al*, 2012; Draber *et al*, 2015), we next assessed the role of HOIP in A20 recruitment to TRAIL-induced signalling complexes. Notably, association of A20 with both, TRAIL

complex I and II, was completely HOIP-dependent (Figs 4A and B, and EV2B and C). LUBAC is constitutively associated with two DUBs, namely OTULIN and CYLD (Fiil et al, 2013; Elliott et al, 2014; Schaeffer et al, 2014; Takiuchi et al, 2014; Draber et al, 2015), both of which are important regulators of immune receptors signalling (Shimizu et al, 2015; Damgaard et al, 2016; Elliott & Komander, 2016). In TNF signalling, CYLD is recruited to the TNFR1-SC via SPATA2 and LUBAC, whereas the linear ubiquitin-specific DUB OTULIN is not, although the latter finding has been controversial (Schaeffer et al, 2014; Draber et al, 2015; Elliott et al, 2016; Kupka et al, 2016; Schlicher et al, 2016; Wagner et al, 2016). We therefore determined whether these DUBs form part of the TRAIL signalling complexes. Whilst CYLD was recruited to TRAIL complex I and to the TRAIL-induced necroptosis-mediating complex in a HOIP-dependent manner, OTULIN did not form part of these complexes (Fig EV2B and C). Interestingly, A20 recruitment to TRAIL complex I was equally restored in A549 HOIP KO by re-expression of HOIP WT or HOIP[AAA], correlating with the accumulation of linear linkages therein, whilst CYLD was not detectable in these conditions probably due to its drastic caspase-dependent cleavage (Fig EV2D).

Thus, HOIP is required for the recruitment of CYLD and A20 to the TRAIL signalling complexes I and II and limits TRAIL-induced apoptosis and necroptosis by modulating the formation and activity of apoptosis- and necroptosis-mediating signalling complexes.

## RIPK1 and caspase-8 are linearly ubiquitinated upon TRAIL stimulation

As LUBAC and linear ubiquitin are detected in the TRAIL complexes I and II, we next aimed to identify LUBAC target(s) in these complexes. To do so, we first used M1-affinity purification (M1-AP) to enrich for M1-ubiquitinated proteins (Draber et al, 2015). Interestingly, both modified RIPK1 and caspase-8 were enriched by M1-AP upon TRAIL stimulation in A549 cells (Fig 5A). To directly assess the presence of M1 chains on RIPK1 and caspase-8, M1-AP samples were treated with viral OTU (vOTU), with or without concomitant OTULIN treatment. Whilst vOTU is a DUB able to cleave all di-ubiquitin linkages except the M1-linkage, OTULIN exclusively hydrolyses the latter (Keusekotten et al, 2013). OTULIN reduced ubiquitination of both, caspase-8 and RIPK1, identifying RIPK1 and caspase-8 as linearly ubiquitinated proteins (Fig 5A). vOTU removed the majority of ubiquitin from RIPK1 and caspase-8 and resulted in the release of free linear ubiquitin chains, as demonstrated by their cleavage in vOTU/OTULIN-cotreated samples (Fig 5A). Thus, the M1 chains on RIPK1 and caspase-8 likely extend from proximal ubiquitin chains of other linkage types, as previously observed for other targets in different signalling complexes (Emmerich et al, 2013). M1-ubiquitin immunoprecipitation (M1-IP) from HOIP-proficient versus HOIP-deficient cells confirmed RIPK1 and caspase-8 as linearly ubiquitinated proteins in response to TRAIL (Fig 5B and C).

## The catalytic activity of HOIP contributes to preventing TRAIL-induced apoptosis but is dispensable for inhibiting necroptosis

To evaluate whether linear ubiquitination contributes to the death-inhibiting function of HOIP, we used HOIP KO K562 cells

reconstituted with empty vector, HOIP WT or the catalytically inactive HOIP C885S (Fig 6A). Contrary to HOIP WT, HOIP C885S failed to prevent TRAIL-induced RIPK1-kinase activity-independent apoptosis (Fig 6B). In line with this, HOIP WT, but not its catalytically inactive version, limited TRAIL-induced cleavage of caspase-8, caspase-3 and PARP-1 (Fig 6C). We next tested whether linear ubiquitination also restricts TRAIL-induced necroptosis by reconstituting HOIP KO MEFs with HOIP WT or C885S (Fig 6D). Interestingly, both limited TRAIL/zVAD-induced cell death to a similar extent (Fig 6E), thereby demonstrating that the activity of HOIP is dispensable for its necroptosis-preventing function. In HOIP-deficient MEFs, TRAIL-induced cell death was completely abrogated by Nec-1s, revealing a RIPK1-kinase-dependent type of cell death. Moreover, these cells did not undergo necroptosis in the absence of zVAD, as evaluated with the RIPK3 inhibitor GSK'872 (Fig 6F). Therefore, HOIP KO MEFs are sensitised to TRAIL-induced RIPK1-kinase-dependent apoptosis. Interestingly, the ubiquitin-ligase activity of HOIP is partially required for protection against this type of cell death (Fig 6E). Thus, HOIP's catalytic activity is dispensable for necroptosis protection whereas it is partially required for preventing TRAIL-induced RIPK1-dependent apoptosis. In addition, linear ubiquitination is essential for preventing RIPK1-independent apoptosis upon TRAIL stimulation.

To understand the molecular mechanism underlying the role of HOIP as a scaffold in the regulation of TRAIL-induced necroptosis, we evaluated the formation of the necroptosis-inducing complex. Expression of HOIP WT, but not HOIP C885S, limited the recruitment of RIPK1 to FADD upon TRAIL/zVAD stimulation. Interestingly, however, the accumulation of RIPK3, which was heavily ubiquitinated in this complex, was limited by both HOIP WT and HOIP C885S, correlating with their ability to prevent TRAIL-induced necroptosis (Fig 6G). Hence, HOIP acts as a scaffold to prevent TRAIL-induced necroptosis.

## HOIP promotes TRAIL-induced gene-activatory signalling and the ensuing cytokine production

Besides triggering cell death, TRAIL can also induce cytokine production via DD-dependent activation of NF-κB and MAPK pathways (Wajant et al, 2000; Harper et al, 2001; Leverkus et al, 2003; Varfolomeev et al, 2005; Trauzold et al, 2006; Tang et al, 2009; Azijli et al, 2013). We therefore investigated whether LUBAC influences TRAIL-induced gene-activatory signalling. Remarkably, TRAIL-induced NF-κB activation was consistently impaired in absence of HOIP in K562, HeLa, HT29 and A549 cells as well as in inducible HOIP KO primary bone marrow-derived macrophages (BMDMs) (Figs 7A and B, and EV3A and B, and EV4B). Moreover, re-expression of HOIP WT in HOIP KO A549, K562 and HT29 cells restored TRAIL-induced NF-κB activation (Figs 7B and C and EV4B). Similar results were observed upon TNF stimulation in HeLa as well as BMDMs (Fig EV3C and D).

In addition to NF-κB, activation of p38, JNK and ERK was consistently impaired in both HOIP KO HeLa and BMDMs in response to TNF (Fig EV3C and D). By contrast, among the cell types tested, absence of HOIP impaired TRAIL-induced p38 and JNK phosphorylation only in HT29 cells (Fig EV4B). Interestingly, HOIP-deficient K562, HeLa and HT29 cells showed an impairment of TRAIL-induced ERK activation (Figs 7A and C, and EV4B), which we

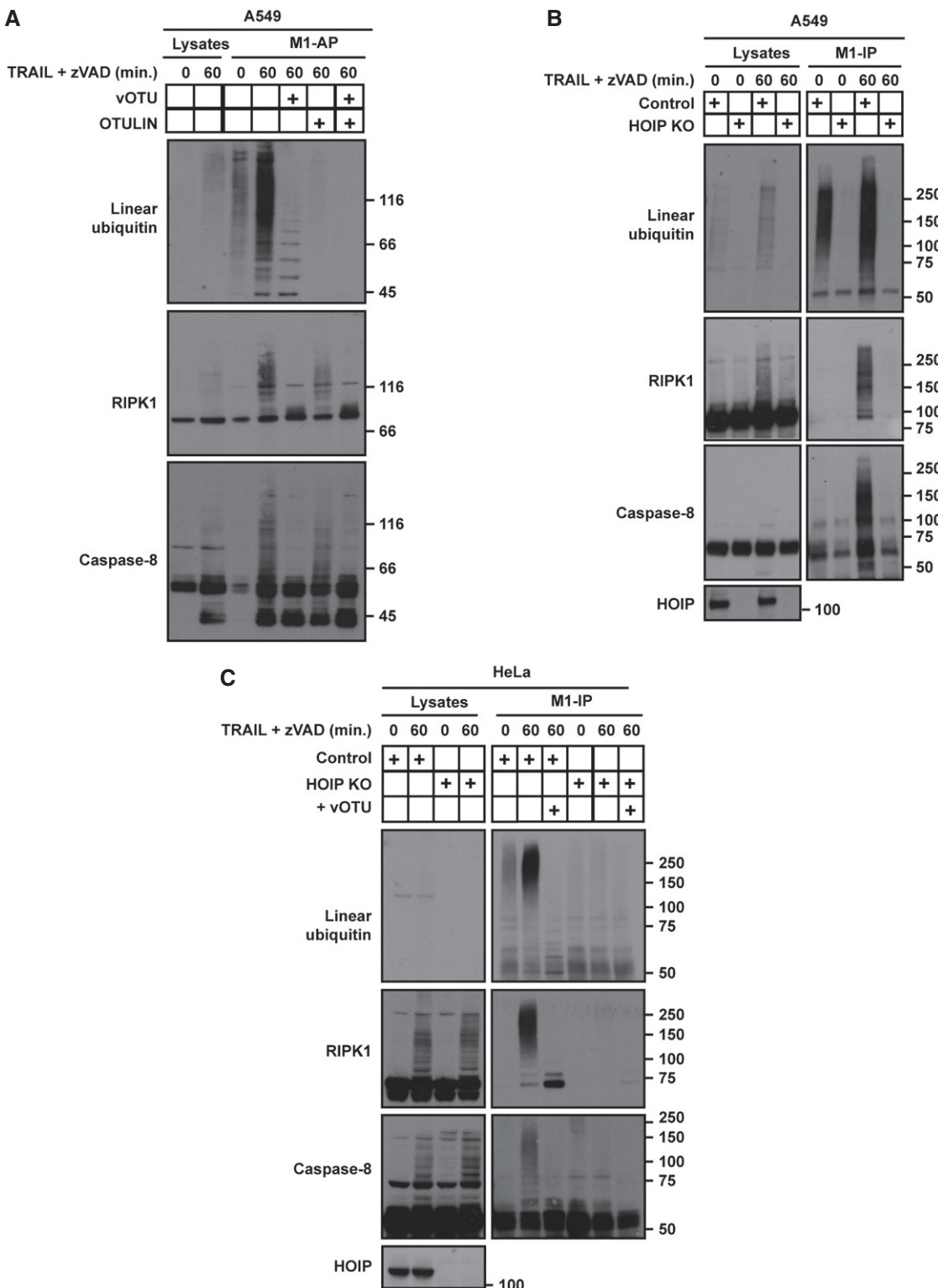

**Figure 5.  RIPK1 and caspase-8 are linearly ubiquitinated upon TRAIL stimulation.**

A   A549 WT cells, pre-treated for 1 h with zVAD, were treated with iz-TRAIL (500 ng/ml) for 1 h and lysed in denaturing conditions. Linear ubiquitin affinity purification (M1-AP) was performed, with subsequent treatment with the indicated DUBs (1 μM), and samples were analysed by Western blot.

B   Control and HOIP KO A549 cells, pre-treated with zVAD for 1 h, were treated with iz-TRAIL (500 ng/ml) for 1 h and lysed in denaturing conditions. Linear ubiquitin immunoprecipitation (M1-IP) was performed, and samples were analysed by Western blot.

C   Control and HOIP KO HeLa cells, pre-treated with zVAD for 1 h, were treated with iz-TRAIL (500 ng/ml) for 1 h and lysed in denaturing conditions. Linear ubiquitin immunoprecipitation (M1-IP) was performed, with subsequent treatment with vOTU (1 μM), and samples were analysed by Western blot.

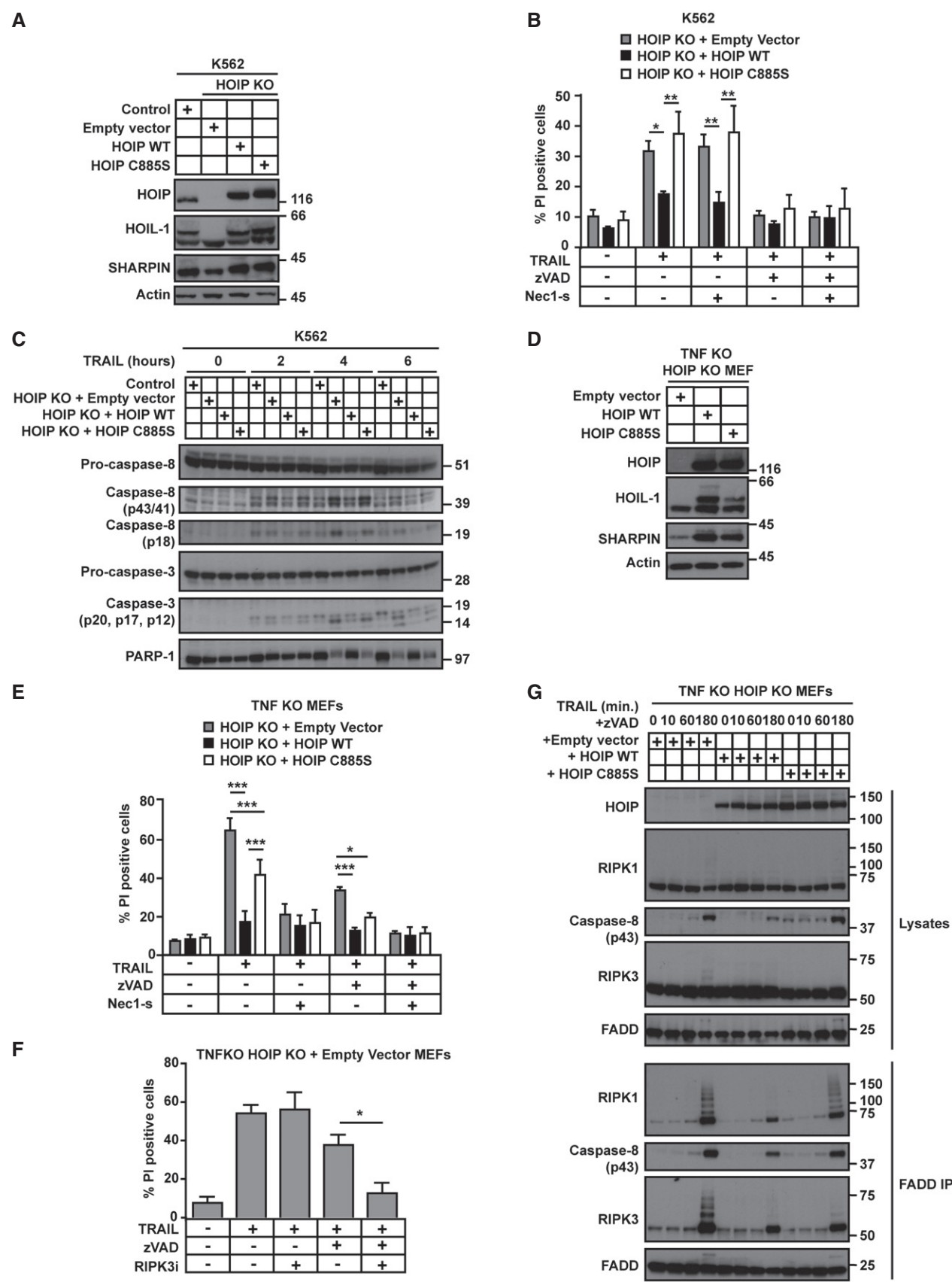

**Figure 6.**

◀

**Figure 6.  The catalytic activity of HOIP contributes to preventing TRAIL-induced apoptosis but is dispensable for preventing necroptosis.**

A   Lysates from control or HOIP KO K562 cells reconstituted with empty vector, HOIP WT or HOIP C885S were analysed by Western blot.

B   HOIP KO K562 cells reconstituted with empty vector, HOIP WT or HOIP C885S, pre-treated with zVAD and/or Nec-1s for 1 h as indicated, were treated with iz-TRAIL (1 μg/ml) for 24 h. Percentage of cell death was determined by flow cytometry after PI labelling (*n* = 5; mean ± SEM). *P < 0.05, **P < 0.01; statistics were performed using ANOVA.

C   Control and HOIP KO K562 cells reconstituted with empty vector, HOIP WT or HOIP C885S were treated with iz-TRAIL (100 ng/ml) for the indicated times. Lysates were analysed with the indicated antibodies.

D   Lysates from TNF KO HOIP KO MEFs reconstituted with empty vector, HOIP WT or HOIP C885S were analysed by Western blot.

E   TNF KO HOIP KO MEFs reconstituted with empty vector, HOIP WT or HOIP C885S, pre-treated with zVAD and Nec-1s for 1 h as indicated, were treated with iz-TRAIL (1 μg/ml) for 24 h. Percentage of cell death was determined by flow cytometry after PI labelling (*n* = 4; mean ± SEM). *P < 0.05, ***P < 0.001; statistics were performed using ANOVA.

F   TNF KO HOIP KO MEFs reconstituted with empty vector, pre-treated with zVAD and/or the RIPK3 inhibitor GSK'872 for 1 h as indicated, were treated with iz-TRAIL (1 μg/ml) for 24 h. Percentage of cell death was determined by flow cytometry after PI labelling (*n* = 3; mean ± SEM). *P < 0.05; statistics were performed using *t*-test.

G   TNF KO HOIP KO MEFs reconstituted with empty vector, HOIP WT or HOIP C885S, pre-treated with zVAD for 1 h, were treated with iz-TRAIL (1 μg/ml) for the indicated times. FADD-containing complexes were immunoprecipitated and analysed by Western blot.

further investigated. Since IKK can trigger activation of MEK1/2 (Gantke *et al*, 2011; Sasaki *et al*, 2013), we hypothesised that TRAIL-induced activation of ERK could be IKK-dependent. Indeed, inhibition of IKK limited TRAIL-induced phosphorylation of IκBα and ERK in K562 cells (Fig EV3E). Moreover, whilst gene activation can suppress TRAIL-induced cell death (Ehrhardt *et al*, 2003; Luo *et al*, 2004; Roué *et al*, 2007; Jennewein *et al*, 2012; Geismann *et al*, 2014; Jeon *et al*, 2015), HOIP KO cells were still sensitised to TRAIL-induced apoptosis in the presence of cycloheximide (Fig EV3F). Our earlier data suggested that TRAIL-induced HOIP cleavage removes its PUB, ZF and NZF1 domains. Since the PUB domain mediates the interaction with CYLD and OTULIN (Schaeffer *et al*, 2014; Takiuchi *et al*, 2014; Draber *et al*, 2015) and the ZF and NZF1 are required for HOIP binding to ubiquitin and NEMO (Haas *et al*, 2009; Ikeda *et al*, 2011; Fujita *et al*, 2014), we hypothesised that cleavage of HOIP may modulate TRAIL-induced gene activation. However, re-expression of HOIP WT or HOIP^AAA equally restored TRAIL-induced activation of NF-κB and MAPKs in HOIP KO cells (Fig EV4A–D). Thus, HOIP consistently promotes activation of NF-κB and, in certain cell types, participates in MAPK activation. However, like the cell-death-limiting function, also the gene-activatory capacity of HOIP is not affected by its caspase-dependent cleavage.

As a functional consequence of impaired gene-activatory signalling, TRAIL-induced production of IL-8 and CCL-2 was largely reduced in both HOIP KO A549 and HeLa cells (Fig 7D and E).

Interestingly, whilst inhibition of IKKα/β, MEK1/2, JNK1/2 and p38α/β all diminished TRAIL- and TNF-induced cytokine production, that of IKKα/β/IκBα/NF-κB was most potent in doing so (Appendix Fig S3A–C). In accordance, KD of p65 consistently reduced IL-8 and CCL-2 production upon both TRAIL and TNF stimulation (Fig 7F and G). Therefore, NF-κB signalling, which is consistently affected by absence of HOIP, is crucial for TRAIL- and TNF-induced pro-inflammatory cytokine production.

**HOIP acts downstream of FADD, caspase-8 and cIAP1/2 to recruit the IKK complex to the TRAIL complex I**

We next aimed to understand how HOIP promotes TRAIL-induced NF-κB activation. Surprisingly, the IKK complex was recruited to the TRAIL-R-associated complex I (Fig 8A), which was so far thought to exclusively act as a DISC. Recruitment of the IKK complex and HOIP to complex I occurred concomitantly and was precisely reflected in the kinetics of degradation of IκBα, a functional consequence of IKK activation. Strikingly, in HOIP KO cells TRAIL-induced recruitment and activation of the IKK complex was not detectable in complex I (Figs 8A and EV5A). In addition, activation of the IKK complex in complex II was also HOIP-dependent (Fig EV5A).

We next wanted to define the sequence of events leading to complex I assembly. Since FADD, caspase-8, RIPK1 and cFLIP_{L/S} regulate TRAIL-induced NF-κB activation (Lin *et al*, 2000;

**Figure 7.  HOIP promotes TNF- and TRAIL-induced gene-activatory signalling and ensuing cytokine production.**                                                                         ▶

A   Control and HOIP KO HeLa cells were stimulated for the indicated times with iz-TRAIL (100 ng/ml), and lysates were analysed by Western blot.

B   HOIP KO A549 cells reconstituted with empty vector or HOIP WT were treated with iz-TRAIL (200 ng/ml) for the indicated times, and lysates were analysed by Western blot.

C   HOIP-deficient K562 cells reconstituted with empty vector or HOIP WT were stimulated for the indicated times with iz-TRAIL (1 μg/ml). Lysates were analysed by Western blot.

D   WT and HOIP KO A549 cells reconstituted with empty vector or with HOIP WT were pre-treated with QVD (10 μM) for 1 h and further treated with iz-TRAIL (100 ng/ml) for 24 h as indicated. IL-8 and CCL-2 concentrations in culture supernatants were measured via ELISA (*n* = 3; mean ± SEM).

E   Control and HOIP KO HeLa cells were pre-treated with QVD (10 μM) for 1 h and further treated with iz-TRAIL (100 ng/ml) or His-TNF (50 ng/ml) for 24 h as indicated. IL-8 and CCL-2 concentrations in culture supernatants were measured via ELISA (*n* = 3; mean ± SEM).

F   WT A549 cells were transfected with siRNA control or targeting p65. 72 h later, cells were pre-treated with QVD (10 μM) for 1 h and further treated with iz-TRAIL (100 ng/ml) or His-TNF (50 ng/ml) for 24 h as indicated. IL-8 and CCL-2 concentrations in culture supernatants were measured via ELISA (*n* = 4; mean ± SEM). Lysates were analysed by Western blot.

G   WT HeLa cells were transfected with siRNA control or targeting p65. 72 h later, cells were pre-treated with QVD (10 μM) for 1 h and further treated with iz-TRAIL (100 ng/ml) or His-TNF (50 ng/ml) for 24 h as indicated. IL-8 and CCL-2 concentrations in culture supernatants were measured via ELISA (*n* = 4; mean ± SEM). Lysates were analysed by Western blot.

Data information: See also Figs EV3 and EV4, and Appendix Fig S3.

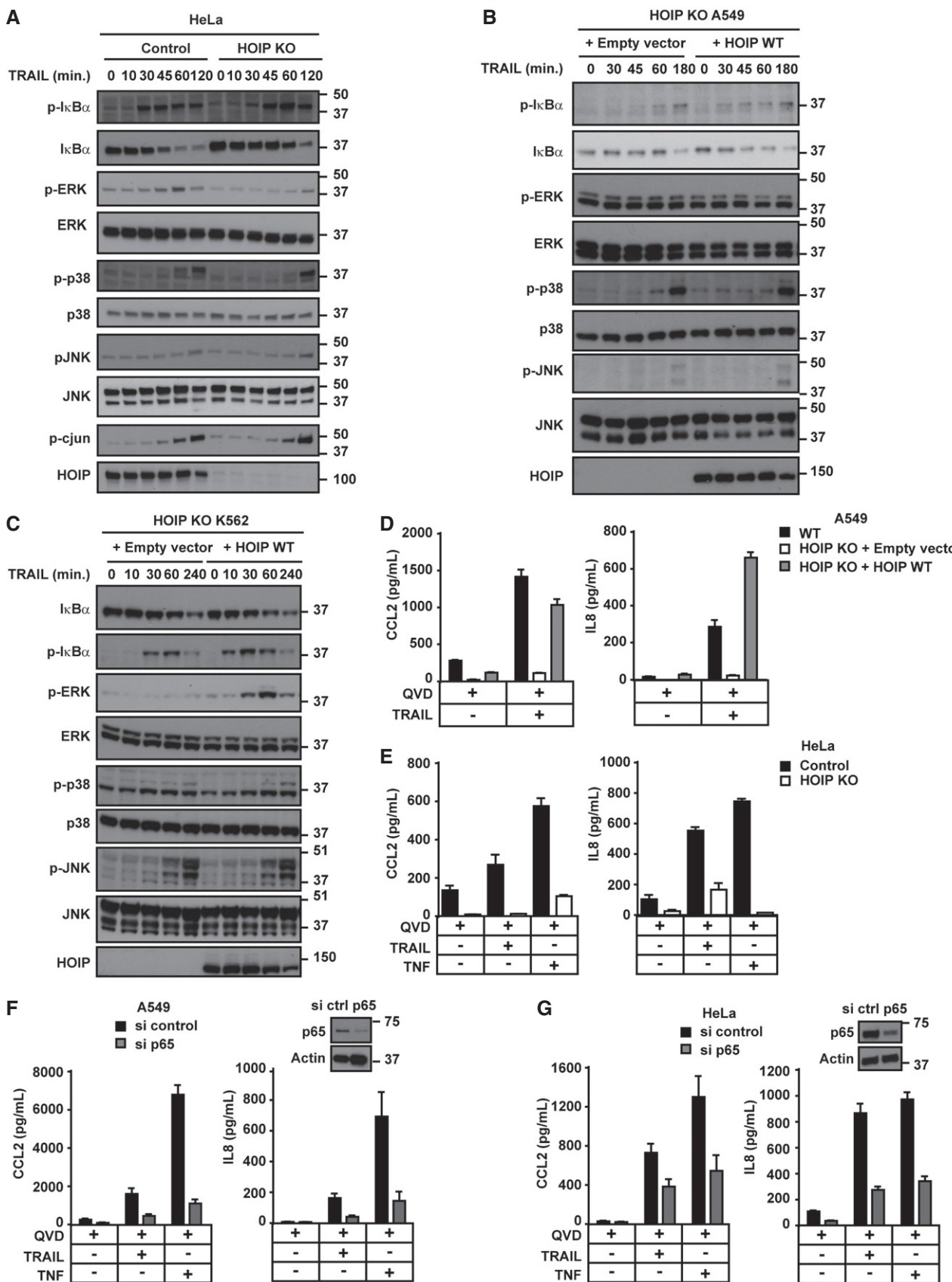

Figure 7.

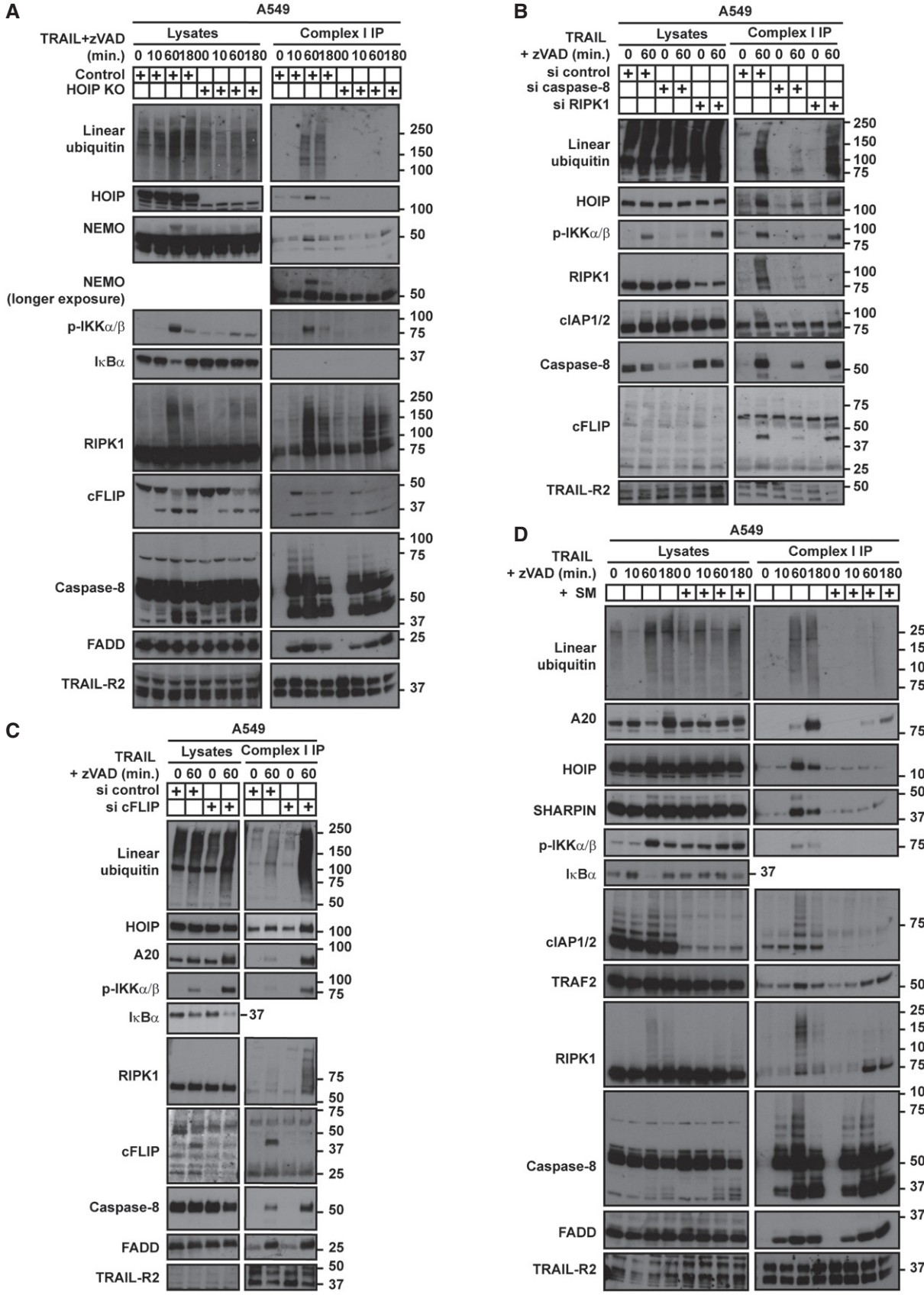

Figure 8.

◄ **Figure 8.  HOIP-mediated recruitment of the IKK complex to TRAIL complex I is caspase-8- and cIAP1/2-dependent, RIPK1-independent and negatively regulated by cFLIP$_{L/S}$.**

A  Control and HOIP KO A549 cells, pre-treated for 1 h with zVAD, were treated with FLAG-lz-TRAIL (500 ng/ml) for the indicated times.

B  A549 WT cells were transfected with siRNA control, siRNA targeting caspase-8 or targeting RIPK1. 72 h later, cells were pre-treated with zVAD for 1 h and treated with FLAG-lz-TRAIL (500 ng/ml) for 1 h.

C  A549 WT cells were transfected with siRNA control or targeting cFLIP$_{L/S}$. 72 h later, cells were pre-treated with zVAD for 1 h and treated with FLAG-lz-TRAIL (500 ng/ml) for 1 h.

D  A549 WT cells, pre-treated with zVAD and SM for 1 h as indicated, were treated with FLAG-lz-TRAIL (500 ng/ml) for the indicated times.

Data information: In all panels, TRAIL complex I was immunoprecipitated via anti-FLAG beads and samples were analysed by Western blot. See also Fig EV5.

Varfolomeev *et al*, 2005; Siegmund *et al*, 2007; Kavuri *et al*, 2011; Grunert *et al*, 2012) and cIAP1/2 recruit LUBAC to various signalling complexes (Haas *et al*, 2009; Varfolomeev *et al*, 2012), we assessed the role of these factors in LUBAC recruitment and IKK activation to complex I. We found that FADD was required for recruitment of caspase-8, cFLIP$_L$, RIPK1, cIAP1/2 and LUBAC to complex I (Figs 2C and EV5B). Furthermore, KD of caspase-8 substantially reduced recruitment of cFLIP$_{L/S}$, RIPK1, cIAP1/2 and HOIP to, as well as activation of IKKα/β in, this complex (Fig 8B). By contrast, silencing of RIPK1 did not affect recruitment of the other complex I components (Fig 8B). Strikingly, depletion of cFLIP$_{L/S}$ resulted in enhanced recruitment of caspase-8, RIPK1, A20 and HOIP to complex I and, accordingly, to increased linear ubiquitination and enhanced phosphorylation of IKKα/β. Notably, A20, a known NF-κB target gene, was strongly induced upon TRAIL stimulation in cFLIP$_{L/S}$ KD cells (Fig 8C). Lastly, depletion of cIAP1/2 by treatment with SMAC mimetics reduced the recruitment of LUBAC and A20 and the accumulation of linear ubiquitin as well as phosphorylation of IKKα/β in complex I whilst recruitment of FADD, caspase-8 and TRAF2 was unaffected (Fig 8D). In summary, LUBAC promotes recruitment of the IKK complex to the TRAIL-R-associated complex I, downstream of FADD, caspase-8 and cIAP1/2, yet independently of RIPK1 (Fig EV5C).

## Discussion

Posttranslational modifications, including phosphorylation and ubiquitination, are major modulators of immune signalling. Here, we provide evidence for LUBAC as a previously unrecognised regulator of TRAIL-induced NF-κB activation, apoptosis and necroptosis by acting in both, complex I and complex II of TRAIL signalling.

Importantly, we establish that TRAIL complex I not only recruits death-mediating factors but also gene-activatory factors, thereby initiating the molecular events leading to gene activation. We identify FADD as essential for recruitment of all complex I components studied. Hence, FADD, which acts as the first interactor with TRAIL-Rs (Kischkel *et al*, 2000; Sprick *et al*, 2000) and is required for TRAIL-induced apoptosis (Walczak *et al*, 1997), serves as the apical factor for all DD-dependent TRAIL signalling outputs. Although cFLIP$_{L/S}$ can interact directly with FADD, caspase-8 depletion decreases its recruitment to complex I, consistent with a previous report showing that its recruitment to complex I is mainly mediated by direct interaction with caspase-8 (Hughes *et al*, 2016). Intriguingly, both FADD and caspase-8 are required for recruitment of RIPK1 to the TRAIL complex I, implying that caspase-8 might stabilise a DD-mediated FADD-RIPK1 interaction. Downstream of

caspase-8, TRAF2 recruits cIAP1/2, promoting recruitment of LUBAC and ubiquitination of RIPK1 and caspase-8. LUBAC ultimately limits caspase-8 activation and recruits the IKK complex, likely due to the ability of NEMO to bind linear chains (Rahighi *et al*, 2009).

In addition to recruiting IKK to complex I and limiting caspase-8 activation, HOIP also mediates recruitment of A20 and CYLD, which might in turn restrict IKK activation, as suggested for TNF signalling (Tokunaga *et al*, 2012; Verhelst *et al*, 2012; Draber *et al*, 2015). In addition, cFLIP$_{L/S}$ acts as a negative regulator of RIPK1, caspase-8 and HOIP recruitment, thereby restricting accumulation of linear ubiquitination and IKK activation in complex I. Importantly, by showing that HOIP mediates recruitment of the IKK complex to complex I of TRAIL signalling, our study extends the current model of TRAIL-induced gene-activatory signalling according to which this type of signalling outcome emanates from complex II (Varfolomeev *et al*, 2005) whilst complex I would be responsible for induction of cell death. According to the revised model for TRAIL signalling (Fig EV5C) and, interestingly, in contrast to TNF signalling, both TRAIL-induced complexes I and II can initiate gene activation as well as cell death with LUBAC playing a crucial role in determining which signalling outcomes emanate from these complexes.

As observed also upon TNF stimulation, absence of HOIP never fully abrogated but rather attenuated and delayed phosphorylation of IκBα. Interestingly, Zhang and colleagues suggested that whilst HOIP might contribute to the early activation of NF-κB, a delayed LUBAC-independent NF-κB activation pathway, which would involve caspase-3 activation and the protein MEKK1, is also induced by TRAIL (Zhang *et al*, 2015). In addition, whilst NEMO preferentially binds to linear ubiquitin chains, it is able to bind K63 chains, albeit with less affinity (Rahighi *et al*, 2009). Therefore, and despite the fact that we did not detect IKK in complex I and II in the absence of HOIP, one could speculate that the presence of K63 chains in these complexes could allow for suboptimal recruitment and activation of IKK, explaining the observed residual IκBα activation. Further work will be needed to precisely define these LUBAC-independent alternative pathways leading to NF-κB activation as well as the molecular mechanisms of their initiation from complex I, II or perhaps additional, yet-to-be-defined TRAIL-induced signalling complexes.

We show that LUBAC is present in the TRAIL-R-associated complex I and the secondary complex II upon TRAIL stimulation and that it forms linear ubiquitin chains in both complexes. We specifically identify RIPK1 and caspase-8 as M1-ubiquitinated proteins following TRAIL stimulation and HOIP's catalytic activity as required for preventing RIPK1-kinase-independent apoptosis. Upon TNF stimulation, linear ubiquitination of RIPK1, but also

TRADD, NEMO and TNFR1, occurs at the TNFR1-SC (Haas *et al*, 2009; Gerlach *et al*, 2011; Draber *et al*, 2015), stabilising this gene-activatory complex and limiting the formation of the death-inducing complex II. Similarly, in TRAIL signalling, RIPK1 association with complex I is reduced and it accumulates in complex II in absence of HOIP. Furthermore, we identify a novel role played by linear ubiquitination in preventing TRAIL-induced apoptosis by discovering that HOIP limits caspase-8 activity in both, complex I and II. It will be interesting to determine whether this function is a direct consequence of linear ubiquitination of caspase-8 or results from an indirect mechanism. Within complex I and II, we found that caspase-8 itself cleaves HOIP. However, this neither abrogated HOIP's cell-death-limiting function nor its ability to enhance gene activation. This could be explained by a sufficient amount of linear ubiquitination already accumulating on components of complex I and II which could promote IKK recruitment and restrict caspase-8 activity before a substantial amount of HOIP would be processed.

Presence but not activity of HOIP is required for prevention of TRAIL-induced necroptosis. The same applies to the association of RIPK3 with RIPK1 and ubiquitination of RIPK3. RIPK3 is also ubiquitinated during TNF-induced necroptosis, which was suggested to support the association of RIPK3 with RIPK1 (Onizawa *et al*, 2015). It remains to be determined whether TRAIL-induced RIPK3 ubiquitination also promotes necroptosis. Yet, according to our results, the E3-ligase(s) and/or DUB(s) responsible for its accumulation are likely to be regulated by HOIP independently of its activity. Given the demonstrated function of CYLD in necroptosis (Hitomi *et al*, 2008; O'Donnell *et al*, 2011), it is unlikely to account for HOIP's protective role in necroptosis despite being recruited to complex I and II together with LUBAC.

In conclusion, LUBAC and M1-ubiquitin chains are previously unrecognised components of complex I and II of TRAIL signalling. Within both complexes, LUBAC limits activation of caspase-8 and promotes IKK complex recruitment, thereby restricting apoptosis and driving pro-inflammatory cytokine production. In addition, LUBAC restricts TRAIL-induced necroptosis by limiting the formation of the necroptosis-mediating complex. It will be interesting to define how particular deubiquitination events in turn modulate TRAIL-induced signalling. In the light of our findings, the addition of SMAC mimetics, which we found limit LUBAC recruitment to complex I, or LUBAC inhibitors, may improve anti-cancer therapeutic effects of TRAIL-R agonists through distinct mechanisms, that is not only by increasing the cancer cell-killing efficiency (Geserick *et al*, 2009; Basit *et al*, 2012; Abhari *et al*, 2013; Fulda, 2014) but also by interfering with the creation of a pro-tumorigenic microenvironment by preventing surviving cancer cells from producing tumour-promoting cytokines.

# Materials and Methods

### Viability and cell-death assays

Cells, pre-treated with zVAD-fmk (20 μM, Abcam), Nec-1s (10 μM, Biovision), cycloheximide (0.5 μg/ml, Sigma), GSK'872 (1 μM, Biovision) for 1 h as indicated, were treated for 24 h with the indicated concentrations of iz-human TRAIL (for HeLa and K562 cells) or iz-murine TRAIL (for MEFs). For cell-death assays, the supernatant was then collected, remaining attached cells were trypsinised, and both were combined and centrifuged at 806 *g* for 5 min. The pellet was resuspended with PBS containing 5 μg/ml propidium iodide (PI) (Sigma). Data were acquired on BD Accuri C6 or alternatively BD LSR Fortessa X20, and the percentage of PI-positive cells was determined by data analysis using FlowJo 7.6.5. For cell viability, Cell Titer Glo (Promega) was used, according to the manufacturer's protocol.

### Cell stimulation, TRAIL complex I, complex II, FADD and HOIP immunoprecipitations

For gene-activatory signalling kinetics, cells seeded in 6-well plates were incubated overnight and stimulated in serum-free medium. For immunoprecipitations, cells were washed with PBS twice and stimulated in serum-free medium. Whenever stated, cells were pre-treated with SM-083 (100 nM) with or without zVAD (20 μM) or TPCA-1 (5 μM, Tocris) for 1 h. For TRAIL complex I IP, cells were stimulated with FLAG-lz-TRAIL as indicated. Cells were lysed in IP-lysis buffer (30 mM Tris–HCl, pH 7.4, 120 mM NaCl, 2 mM EDTA, 2 mM KCl, 10% glycerol, 1% Triton X-100, 1× COMPLETE protease-inhibitor cocktail and 1× PhosSTOP (Roche)) at 4°C for 30 min. Lysates were cleared by centrifugation at 17,000 *g* for 30 min. FLAG-lz-TRAIL (200 ng) was added to the non-stimulated samples before all samples were pre-cleared using Sepharose beads (Sigma) for 1 h at 4°C. 15 μl of M2 beads (Sigma) were then added to the samples and incubated overnight at 4°C. To analyse the complex II, the complex I-depleted lysates were collected and incubated overnight at 4°C with 15 μl protein G beads pre-blocked with 1% BSA and coupled with 3 μg anti-caspase-8 antibody (Santa Cruz Biotechnology, clone C20). For FADD IP, cells were stimulated with iz-murine TRAIL and zVAD as indicated and lysates were prepared as described for the TRAIL complex I IP. 15 μl of protein G beads pre-blocked in 1% BSA and coupled with 3 μg anti-murine FADD antibody (Santa Cruz Biotechnology, clone M19) were added to the supernatants and incubated overnight at 4°C. For HOIP IP, cells expressing HOIP-TAP or empty vector were stimulated with iz-human TRAIL and zVAD as indicated and samples were processed as indicated for TRAIL complex I IP. After all IPs, beads were washed 4 times with IP-lysis buffer and incubated with LDS containing 5 mM DTT at 95°C for 5 min before Western blot analysis.

### Isolation of linearly ubiquitinated proteins by immunoprecipitation (M1-IP) and affinity purification (M1-AP)

For M1-IP, cells were lysed in M1-IP lysis buffer (5 M urea, 135 mM NaCl, 1% Triton X-100, 1.5 mM MgCl$_2$, 2 mM N-ethylmaleimide, 1% SDS, 1× COMPLETE protease-inhibitor and 1× PhosSTOP (Roche)). Lysates were incubated 20 min on ice, sonicated and cleared by centrifugation at 17,000 *g* for 30 min. Lysates were pre-cleared with Sepharose beads (Sigma) and incubated with 0.25 μg antibody per sample (linear ubiquitin antibody, clone 1E3, Millipore) overnight at room temperature. Protein G beads (GE Healthcare) were added for 2 h, and beads were washed twice with M1-IP lysis buffer and twice with PBS before performing DUB assay.

For M1-AP, cells were lysed in AP-lysis buffer (30 mM Tris–HCl, pH 7.4, 120 mM NaCl, 2 mM EDTA, 2 mM KCl, 0.5% CHAPS, 1% SDS, 1× COMPLETE protease-inhibitor and 1× PhosSTOP (Roche)).

Lysates were incubated 10 min on ice, sonicated and cleared by centrifugation at 17,000 *g* for 30 min. The M1-AP tool was produced as described previously (Draber *et al*, 2015). Samples were diluted to 0.1% SDS before M1-ubiquitin-specific recombinant affinity protein freshly pre-coupled to HALO beads (Promega) was added for overnight incubation at 4°C. Beads were washed three times using AP-lysis buffer deprived of SDS, proteases and phosphatases inhibitors before undergoing DUB assay.

### Deubiquitination assay

Recombinant viral OTU and OTULIN were produced as described previously (Draber *et al*, 2015). After the washes, beads were resuspended in DUB reaction buffer (50 mM Tris (pH 7.5), 50 mM NaCl and 5 mM DTT) with or without 1 μM recombinant deubiquitinase and incubated for 1 h at 37°C. Reaction was stopped by adding LDS (Invitrogen) with 5 mM DTT, and samples were reduced and denatured by incubation for 10 min at 70°C before Western blot analysis.

### Caspase assay

K562 cells were lysed in caspase assay lysis buffer [30 mM Tris–HCl, pH 7.4, 120 mM NaCl, 2 mM EDTA, 2 mM KCl, 10% glycerol, 1% Triton X-100, 1× PhosSTOP (Roche), 5 mM DTT, AEBSF 70 μM and pepstatin A (1 μM)], at 4°C for 30 min. Lysates were cleared by centrifugation at 17,000 *g* for 30 min. 3 U of active caspases 7, 6 (Enzo Life Sciences), 3, 8 or 10a (produced by Martin Sprick) was added to the cleared lysates and incubated for 2 h at 37°C. The reaction was stopped by adding LDS (Invitrogen) with 5 mM DTT, and samples were reduced and denatured by incubation for 10 min at 70°C before Western blot analysis.

TAP-HOIP was immunoprecipitated from K562-HOIP-TAP expressing cells as described earlier. After 4 washes in IP-lysis buffer, the beads were resuspended with caspase assay buffer (20 mM HEPES, pH 7.4, 0.1% CHAPS, 5 mM DTT, 2 mM EDTA, 5% sucrose) containing 3 U of recombinant caspases 7, 6, 3, 8 or 10a and incubated for 2 h at 37°C. The reaction was stopped by adding LDS (Invitrogen) with 5 mM DTT, and samples were reduced and denatured by incubation for 10 min at 70°C before Western blot analysis.

### Electrophoresis and Western blot

Proteins were separated using 4–12% Bis–Tris–NuPAGE gels (Invitrogen) with NuPAGE® MOPS running buffer. Alternatively, 4–15% Mini-PROTEAN® TGX™ Precast Protein Gels and TGX buffer (Bio-Rad) were used. Proteins were transferred from gels onto ECL-Membrane Hybond 0.45-μm nitrocellulose membrane (GE Healthcare). Alternatively, the Trans Blot® Turbo™ System (Bio-Rad) was used. Whenever necessary, 50 mM glycine, pH 2.3 was used as stripping buffer in between antibody incubations.

### ELISA

A549 and HeLa cells were pre-treated with QVD-OPh (10 μM, Abcam) for 1 h, with or without TPCA-1 (10 μM), SP600125 (15 μM), losmapimod (5 μM) or PD184352 (1 μM), and further stimulated with iz-human TRAIL (100 ng/ml) for 24 h as indicated. Medium was collected and centrifuged at 405 *g* for 3 min. IL-8 and CCL2 concentrations were determined in the resulting supernatants via ELISA obtained from R&D, according to the manufacturer's instructions.

### Production of recombinant TRAIL and TNF

Iz-human TRAIL and iz-murine TRAIL were produced and purified as described previously (Ganten *et al*, 2006). The sequences coding for FLAG-lz-TRAIL, consisting of a His-tag followed by 1xFLAG, a PreScission cleavage site, 2xStrep-tag II, a leucine zipper motif and part of the extracellular portion of human TRAIL (aa 120–281) were inserted into the pQE30 vector. FLAG-lz-TRAIL was then produced and purified as described previously for TAP-TNF (Draber *et al*, 2015). Iz-human TRAIL, iz-murine TRAIL and FLAG-lz-TRAIL were tested as LPS-free using Pierce LAL Chromogenic Endotoxin Quantitation Kit (Thermo Scientific). His-TNF was produced as described previously (Draber *et al*, 2015).

### siRNA-mediated knock-down

All siRNAs used were ON TARGET Plus SMART pool from Dharmacon, non-targeting, targeting cFLIP$_{L/S}$, caspase-8, RIPK1 or p65. A549 WT cells were reverse-transfected in 10-cm dishes, using 14 μl of Dharmafect I (GE Healthcare)/dish and 40 nM siRNA pools. Cells were stimulated 72 h after transfection.

### Cell lines and antibodies

WT cell lines were purchased from ATCC. HOIP-deficient HT29 cells were generated by transfection with pSpCas9(BB)-2A-GFP (PX458, Addgene plasmid #48138) (Ran *et al*, 2013) containing the following gRNA-encoding sequence targeting exon 2 of RNF31: CACCGTTGA CACCACGCCAGTACCG. GFP-positive cells were sorted into 96-wells plates, and single-cell clones were analysed 4 weeks later by Western blot to assess KO efficiency. All the other KO and reconstituted cell lines used were described previously (Peltzer *et al*, 2014; Draber *et al*, 2015; von Karstedt *et al*, 2015). For reconstitution of HOIP KO cells with HOIP WT and HOIPD348A, HOIPD387A and HOIPD348A/D387A/D390A, the different versions of HOIP were generated by site-directed mutagenesis and inserted into pBabe-puro (Addgene) and Phoenix-Ampho cells were transfected. A549, HT29 and K562 HOIP KO cells were infected with the obtained viral supernatants and then selected with puromycin for 4 weeks. All cell lines were regularly tested for mycoplasma using MycoAlert™ Mycoplasma Detection Kit (LONZA). For the isolation of BMDMs, 4-hydroxytamoxifen (4-OHT)-inducible CreERT2$^+$ *Hoip*$^{fl/fl}$ mice were sacrificed. After removal of the hindlimbs, muscle tissue was carefully removed from femur and tibia. Both bones were cut on their extremes, and the bone marrow content was flushed out with PBS using a 25-G needle. The obtained cells were passed through a cell strainer and washed with PBS. Erythrocytes were lysed by resuspension in 2 ml of red blood cell lysis buffer (eBioscience) and incubation for 1 min. Finally, cells were washed and resuspended in conditioned medium (RPMI medium containing 20% foetal calf serum (FCS), 1% penicillin/streptomycin (Invitrogen) and 20% conditioned medium from L929 cells) and seeded in 6-well plate.

The cells were incubated for 7 days before the experiment, replacing the conditioned medium every 2 days. To induce deletion of the HOIP gene, tamoxifen was added 3 days before the experiment to half of the wells at a final concentration of 1 μM. In addition, Enbrel (50 μg/ml, Amgen) was also added to all the wells to avoid autocrine stimulation by TNF. This medium was replaced daily until the experiment was performed.

The following primary antibodies were used for Western blot: cleaved murine caspase-8 (Cell Signaling, 9429), cleaved caspase-3 (Cell Signaling, 9664), caspase-3 (Cell Signaling, 9668), caspase-10 (MBL, M059-3), murine caspase-8 (Enzo Life Sciences, ALX-804-447-C100), caspase-8 (Enzo Life Sciences, C15), cFLIP$_{L/S}$ (Enzo Life Sciences, NF6), Bid (Cell Signaling, 2002), TRAF2 (Enzo Life Sciences, ADI-AAP-422), FADD (Enzo Life Sciences, 1F7), TRAIL-R2 (Cell Signaling, 3696), TRAIL-R1 (ProSci, 1139), FADD (BD, 556402), RIPK1 (BD, 610459), cIAP1/2 (R&D, MAB3400) FADD (Santa Cruz Biotech, sc-5559), PARP-1 (BD, 556362), HOIL-1 (Haas *et al*, 2009), SHARPIN (Proteintech, 14626-1-AP), A20 (Santa Cruz Biotech, sc-166692), HOIP (Aviva Systems Biology, ARP43-241_P050), linear ubiquitin (Millipore, MABS199), p-MLKL (Abcam, ab196436), RIPK3 (Novus, IMG-5846A), murine RIPK3 (Enzo Life Sciences, ADI-905-242-100), NEMO (Santa Cruz Biotech, sc-8330), p-IKK α/β (Cell Signaling, 2697), p-IκBα (Cell Signaling, 9246), IκBα (Cell Signaling, 9242), p105 (Cell Signaling, 4717), p-p105 (Cell-Signaling, 4806), p-MEK1/2 (Cell Signaling, 9154), p-ERK1/2 (Cell Signaling, 4370), ERK1/2 (Cell Signaling, 4695), p-JNK (Cell Signaling, 4671), JNK (Cell Signaling, 9258) p-p38 (Cell Signaling, 9215), p38 (Santa Cruz Biotech, sc-728), Flag (Sigma, M2), actin (Sigma, A1978), p-c-Jun (Cell Signaling, 9261), c-Jun (Cell Signaling, 9165), p-CREB (Cell Signaling, 9198), CREB (Cell Signaling, 9197), HOIP (R&D, AF8039), CYLD (Santa Cruz Biotech, sc-74435), OTULIN (Abcam, ab151117).

**Statistics**

GraphPad Prism 6 (GraphPad software version 6.01) was used for data analysis. Statistical significance between groups was evaluated by two-sided unpaired *t*-test and a two-way ANOVA test whenever performing multiple comparisons, as indicated. Alternatively, and as indicated, a Mann–Whitney *U*-test was used. Whenever relevant, the assumptions of normality and equality of variance were verified using the Shapiro–Wilk test and the *F*-test, respectively. *P*-values < 0.05 are indicated and considered as statistically significant.

**Expanded View** for this article is available online.

## Acknowledgements

We thank all members of the Walczak group for useful technical advice and fruitful scientific discussions. The SMAC mimetic compound SM083 (also known as SM9a) was synthesised and kindly provided by P. Seneci and L. Manzoni. BAX/BAK-DKO HCT-116 cells were kindly provided by B. Vogelstein and R. Youle. This work was supported by a Wellcome Trust Senior Investigator Award (096831/Z/11/Z), an ERC Advanced Grant (294880) and a Cancer Research UK programme grant (A17341) awarded to H.W., a BBSRC CASE studentship (BB/J013129/1) awarded to M.R. and a Dutch Cancer Society (KWF) fellowship (BUIT 2015-7526) awarded to M.S. This work was also supported by the CRUK–UCL Centre grant (515818), the Cancer Immunotherapy Accelerator (CITA) Award (525877) and the National Institute for Health Research University College London Hospitals Biomedical Research Centre from CRUK.

## Author contributions

HW conceived the project. EL and HW designed the research and wrote the manuscript. EL, CK-M, PD, DDM, TH, MR, SK and YS performed experiments. LT and MS helped establish knockout cell lines. MRS provided active caspase-3, caspase-8 and caspase-10a.

## Conflict of interest

H.W. is co-founder and shareholder of Apogenix AG. Otherwise the authors declare that they have no conflict of interest.

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
