## [Review Process File · The EMBO Journal]

Manuscript EMBO-2016-95699

The Linear Ubiquitin Chain Assembly Complex regulates TRAIL-induced gene activation and cell death

Elodie Lafont, Chahrazade Kantari-Mimoun, Peter Draber, Diego De Miguel, Torsten Hartwig, Matthias Reichert, Sebastian Kupka, Yutaka Shimizu, Lucia Taraborrelli, Maureen Spit, Martin R. Sprick and Henning Walczak

Corresponding author: Henning Walczak, University College London

Review timeline:

Submission date:	21 September 2016
Preliminary Editorial Decision:	01 November 2016
Confirmed Editorial Decision:	07 November 2016
Revision received:	30 January 2017
Accepted:	13 February 2017

Editor: Hartmut Vodermaier

Transaction Report:

Preliminary Editorial Decision

01 November 2016

Thank you again for your patience during our external evaluation of your study on LUBAC roles in TRAIL signaling. I would like to once more apologize for the delay in getting back to you with decision. Unfortunately, one of the three referees who agreed to review your manuscript has still not returned a report, despite repeated reminders sent from our offices and affirmations that we would still receive comments shortly. To prevent unnecessary further loss of time, and given that the other two reviewers are overall in fair agreement, I therefore decided to forward you the first two reports at this stage, together with an invitation to start revising the manuscript according to these comments.

As you will see, both referees appreciate the interest of establishing LUBAC roles in TRAIL signaling pathways, but especially referee 1 feels that not all of these findings are conceptually highly unexpected and that some of the more novel aspects would need to be fleshed out in some more detail in order to make this study a compelling candidate for The EMBO Journal. In addition, both referees raise a number of more specific experimental and textual issues that would need to be addressed as well. Should you be able to satisfactorily answer to both of these reviewers (during a revision period in which publication of competing manuscripts elsewhere will have no negative impact on our final assessment of your own study), then we should be happy to consider your manuscript further for eventual publication.

Please note that this formally remains a preliminary decision while the third report is still outstanding. Should we eventually receive these comments within the next days, I would forward

them to you so you could also take them into account. However, please rest assured that only in case referee 3 should bring up major potentially confounding concerns would we still mandate addressing them as well. In any case, I shall be in touch shortly for confirming this preliminary revision invitation.

REFEREE REPORT

Referee #1:

This study by Lafont et al. sets out to determine if the LUBAC ubiquitin E3 complex is involved in TRAIL-induced signalling (cell death and gene activation). Using an extensive array of biochemical, molecular and genetic approaches in cell lines, the author demonstrate that LUBAC is part of the TRAIL-induced receptor complex as well as the soluble, non-receptor bound complex II. Herein, LUBAC generates linear ubiquitin chains (suggestively on RIPK1 and caspase-8) that are needed for recruitment of the deubiquitylating enzyme A20. Interestingly, the authors noted that HOIP was cleaved after TRAIL stimulation, suggestively in a caspase-dependent manner. Functionally, the study shows that LUBAC protects against TRAIL-induced cell death (apoptosis and necroptosis) and that it also contributes to inflammatory signalling and production of the chemokine CCL2 (when cell death is inhibited).

The study is technically of high quality and the data clearly demonstrate a role for LUBAC in regulating TRAIL signalling outcomes in cultured cell lines. However, the role of LUBAC in immune receptor signalling has been extensively studied (by the authors and other groups), including in context of the TNF receptor complex where very similar complexes are formed and where LUBAC has similar functions (e.g. Haas et al. 2009, Gerlach et al. 2011, Draber et al. 2015, etc.). As such, a significant portion of the study essentially confirms what is already known about LUBAC in similar receptor signalling systems. That said, the study contains unexpected and potentially interesting observations that could bring novel insight to LUBAC biology (e.g. that HOIP is cleaved by caspases and that HOIP appears to be selectively required for TRAIL-induced ERK activation). However, the characterisation of these observations is very limited, which leaves the impression that the study is a bit preliminary.

Comments

The authors conclude that HOIP is cleaved by caspase-8 and speculate that this might remove the N-terminal region of HOIP where the deubiquitylating enzymes OTULIN and CYLD bind, thereby affecting the regulation of ubiquitin-dependent processes. However, there does not appear to be any evidence that HOIP is cleaved by caspase-8 apart from the observation that the cleavage is inhibited by zVAD (Figure 2A). A separate study (Joo et al. MCB 2016, published after the submission of this manuscript) similarly finds that HOIP is cleaved in response to TNF and TRAIL treatment. In vitro caspase cleavage assays suggest that caspase-3/6 but not caspase-8 is responsible for HOIP processing. A more comprehensive analysis should be carried out of the observed HOIP cleavage and of the functional implications for cell death and gene activation. This would significantly strengthen the study.

In addition to recruiting A20 to receptor complexes, HOIP also interact with CYLD and OTULIN. However, there is no investigation of the recruitment of CYLD and OTULIN to the TRAIL receptor complexes. This needs to be investigated, in particular because HOIP appears to prevent necroptosis independently of its catalytic activity and CYLD regulates necroptosis after TNF treatment. Additionally, it would be of interest to know if the cleavage of HOIP regulates the association of these deubiquitylases with the signalling complexes.

It is interesting that HOIP seems to primarily regulate ERK activation after TRAIL treatment, while having very limited effects on other MAP kinases and I κ B turnover (Figure 6). However, this is only shown in one cell line and needs to be demonstrated also in other cell types used in the study. Additionally, it will be important to determine the functional consequence of decreased ERK vs

JNK activation on transcriptional responses/cytokine production. In Figure 6, the role of HOIP on ERK activation was determined in K562 cells while its role in CCL2 production was determined in A549 cells, which prevents any conclusion as to functional importance of ERK activity for CCL2 production.

Previously reports indicate that impairment of LUBAC function strongly decreases JNK activation during TNF and CD40L signalling (Haas et al. 2009, Gerlach et al. 2011). Are there differences between TRAIL vs TNF/CD40L that determine this differential requirement for LUBAC in activation of MAPKs? Alternatively, might the observed differences simply be due to cell type differences and not as such differences in TNF and TRAIL signalling? The authors should compare MAP kinase pathways after TRAIL and TNF signalling side-by-side in the same cell lines to resolve this.

Specific comments:

The claim that the HOIP catalytic activity is "required to prevent TRAIL-induced apoptosis" is not fully supported by the experimental data in Figure 5: In K562 cells, reintroduction HOIP WT partially rescued the cell death phenotype and in MEFs catalytically dead HOIP provided partial protection from TRAIL. Please rephrase the statement to reflect the data.

Page 1: The acronym "DD" in "TRAIL also triggers DD-dependent...." Should be explained when first used.

Figure 1: Treatment times should be indicated in the figure legend instead of only in the methods section.

Figure 2 and 3A: It is unclear how/if the "DISC-IP" procedure differs from the "Complex I IP" shown in Figures 6 and 7? The DISC-IP procedure is not described in the M&M section.

On Page 7 the authors refer to their article (Draber et al. 2015) showing that A20 is recruited to the TNFR1-SC via linear ubiquitin. The original papers showing that A20 is recruited to receptor complexes via linear ubiquitin should also be acknowledged: Verhelst et al. 2012 EMBOJ, Tokunaga et al. 2012 EMBOJ.

Figure 5C: Pretreatment with Nec1-s before TRAIL should be included to assess the contribution of necroptosis in MEFs reconstituted with HOIP variants. Without this control it cannot be concluded that HOIP inhibits necroptosis independently of its activity.

Referee #2:

The relevance of the LUBAC and its activity has been intensively studied in recent years in context of TNFR1. However, there is only one study (without in depth analysis) so far (Zhang et al., Cell Signal. 2015 Feb;27(2):306-14) which addressed the role of the HOIP/LUBAC in signaling by the TNFR1-related TRAIL death receptors. The manuscript by Lafont fills this gap and demonstrates i) that the LUBAC is recruited to the plasma membrane associated TRAILR2 complex and is also part of the secondarily formed TRAIL induced cytoplasmic complex II, ii) that the LUBAC inhibits cell death induction by TRAIL but promotes TRAIL-induced NFkB/ERK signaling and iii) that LUBAC recruitment to the TRAILR2 signaling complex occurs downstream of FADD, caspase-8 and cIAPs and is required for bridging the TRAILR2 signaling complex with the IKK complex.

TNFR1 and the TRAIL death receptors share a death domain but based on the published data available so far, they differ considerably in the signaling complexes that are formed on the DDs after receptor stimulation. TNFR1 efficiently recruits TRADD, RIP, TRAF2, the LUBAC and the IKK complex but shows no recruitment of FADD and caspase-8. In contrast, TRAILR2 efficiently recruits FADD and caspase-8 but shows only a poor recruitment of TRAF2 and RIP. The study of Lafont et al, now modifies this picture and not only demonstrates significant recruitment of TRAF2 and RIP1 to TRAILR2 but also that this enables co-recruitment of the LUBAC and the IKK

complex. The data are therefore not only novel but also of broad and general interest for the death receptor field.

I have only a few minor concerns/comments on the manuscript:

Figure 2A - The receptor bands are labeled with TRAILR1. The appearance of the bands and the text suggest, however, that TRAILR2 is shown.

Paragraph "LUBAC is recruited" - The second part of the first sentence "As HOIP prevents TRAIL-induced cell deathas this is the receptor-associated protein complex we and others previously showed initiates this signaling outcome. " is incomprehensible. I suggest rephrasing or deletion.

Figure 3B - The FADD IP of non-stimulated cells shows significant amounts of RIP and RIP3. It should be clarified whether this is due to constitutive FADD-RIP-RIP3 complex formation or whether this represents non-specific binding to the protein G beads.

TRAILR2 and complex II IPs performed with non-stimulated cells frequently contain TRAF2 and RIP. I assume that this represents non-specific binding to the protein G beads and not specific association. This should be resolved for the reader by showing control IPs with protein G beads only versus TRAILR2 and complex II IPs.

Figure EV1 F - A549 cells should be included here as this cell type is used in most experiments in the manuscript.

A scheme of TRAILR2 signaling summarizing the findings of the manuscript could be helpful for the reader.

Figure 6D and 7 - There is significant processing of FLIP-L and caspase-8 in the IPs although TRAIL + the caspase inhibitor ZVAD have been used for stimulation. This should be commented/explained. As DISC-associated caspase activity is obviously not fully blocked, the authors cannot conclude in the discussion that caspase-8 activity has no relevance for LUBAC recruitment etc.

It should be indicated throughout the manuscript whether FLIP-L or FLIP-S or both are meant when the term FLIP is used.

Discussion last paragraph - "As a consequence of TRAIL-induced gene activation several cytokines are produced (Tang et al. 2009; Trauzold et al., 2006; Varfolomeev et al., 2005)." I recommend to cite here earlier papers showing this already many years before (e.g. Harper et al., J Biol Chem. 2001 Sep 14;276(37):34743-52 or Wajant et al., J Biol Chem. 2000 Aug 11;275(32):24357-66).

The authors should discuss Zhang et al., (Cell Signal. 2015 Feb;27(2):306-14) which analyzed TRAIL-induced NFkappaB signaling in HOIP deficient A20 cells and observed a HOIP-dependent and a HOIP-independent mode of NFkappaB signaling!

Confirmed Editorial Decision

07 November 2016

I am writing to formally confirm our earlier preliminary decision on your manuscript, which contained an invitation to revise the manuscript based on reviewer comments I forwarded. Since we have had no further communications from the outstanding third reviewer, you will only need to respond to/address the issues raised by reviewers 1 and 2.

Point-by-point response to the reviewers' comments on manuscript EMBOJ-2016-95699 by Lafont *et al.* entitled:**“The Linear Ubiquitin Chain Assembly Complex regulates TRAIL-induced gene activation and cell death”**

Referee #1:

Comments to the authors (in italics):

This study by Lafont et al. sets out to determine if the LUBAC ubiquitin E3 complex is involved in TRAIL-induced signalling (cell death and gene activation). Using an extensive array of biochemical, molecular and genetic approaches in cell lines, the author demonstrate that LUBAC is part of the TRAIL-induced receptor complex as well as the soluble, non-receptor bound complex II. Herein, LUBAC generates linear ubiquitin chains (suggestively on RIPK1 and caspase-8) that are needed for recruitment of the deubiquitylating enzyme A20. Interestingly, the authors noted that HOIP was cleaved after TRAIL stimulation, suggestively in a caspase-dependent manner. Functionally, the study shows that LUBAC protects against TRAIL-induced cell death (apoptosis and necroptosis) and that it also contributes to inflammatory signalling and production of the chemokine CCL2 (when cell death is inhibited).

The study is technically of high quality and the data clearly demonstrate a role for LUBAC in regulating TRAIL signalling outcomes in cultured cell lines. However, the role of LUBAC in immune receptor signalling has been extensively studied (by the authors and other groups), including in context of the TNF receptor complex where very similar complexes are formed and where LUBAC has similar functions (e.g. Haas et al. 2009, Gerlach et al. 2011, Draber et al. 2015, etc.). As such, a significant portion of the study essentially confirms what is already known about LUBAC in similar receptor signalling systems. That said, the study contains unexpected and potentially interesting observations that could bring novel insight to LUBAC biology (e.g. that HOIP is cleaved by caspases and that HOIP appears to be selectively required for TRAIL-induced ERK activation). However, the characterisation of these observations is very limited, which leaves the impression that the study is a bit preliminary.

Our reply (in blue):

We thank the reviewer for appreciating the technical quality and importance of this study. Whilst the role of LUBAC has been studied in other immune receptor signalling system, our study provides an extensive analysis of its role in TRAIL-induced apoptosis, necroptosis and gene activation. Importantly, TNF and TRAIL signalling are in fact rather different in several aspects: signalling complex formation and composition, sequence of molecular events and biological outcomes (N.B. this notion is also referred to and acknowledged by the second reviewer in their opening statement on the previous version of our manuscript).

Whilst TNF-induced molecular events leading to gene activatory signalling are rather well defined, they remain particularly elusive for TRAIL signalling. Most importantly, we provide first evidence

that HOIP, HOIL-1, SHARPIN, CYLD, A20, RIPK1, TRAF2, cIAP1/2 and the IKK complex are *bona fide* components of the TRAIL-R-associated complex I, in addition to the well-known components TRAILR-1/2, FADD, caspase-8/10 and cFLIP. Notably, this defines TRAIL signalling complex I as a multi-tasking complex, able to induce both gene-activatory signalling and cell death, as opposed to the solely gene-activatory TNFR1-SC. In addition, we demonstrate that LUBAC is a main molecular component switching the signalling outcome from death to gene activation which emanates from TRAIL signalling complex I and II.

We greatly appreciate the constructive points raised by the reviewer, in particular regarding further investigation of the molecular mechanism leading to HOIP cleavage as well as its functional relevance towards apoptosis and gene-activatory signalling. We have also further documented the role of HOIP in kinase pathway activation and ensuing cytokine production in TRAIL and TNF signalling, as suggested. Finally, and as detailed below, we have performed additional experiments to address the reviewer's specific points.

Comments

The authors conclude that HOIP is cleaved by caspase-8 and speculate that this might remove the N-terminal region of HOIP where the deubiquitylating enzymes OTULIN and CYLD bind, thereby affecting the regulation of ubiquitin-dependent processes. However, there does not appear to be any evidence that HOIP is cleaved by caspase-8 apart from the observation that the cleavage is inhibited by zVAD (Figure 2A). A separate study (Joo et al. MCB 2016, published after the submission of this manuscript) similarly finds that HOIP is cleaved in response to TNF and TRAIL treatment. In vitro caspase cleavage assays suggest that caspase-3/6 but not caspase-8 is responsible for HOIP processing. A more comprehensive analysis should be carried out of the observed HOIP cleavage and of the functional implications for cell death and gene activation. This would significantly strengthen the study.

As we completely agree with the reviewer that a more comprehensive analysis of the observed TRAIL-induced HOIP cleavage and the functional implications thereof would significantly strengthen the study we added these analyses to our revised study.

We now included an evaluation of the involvement of caspases 3, 6, 7, 8 and 10 in both, TRAIL- and TNF-induced HOIP cleavage (Figures 3A-C and Appendix Figure S2A-C). We have also commented on our previous data in light of these new findings. As likewise suggested, we have investigated the influence of HOIP cleavage on its apoptosis-preventing function. These experiments form part of the Figures 3 and Appendix Figure S2 and are described in the new paragraph entitled "Caspase 8-dependent cleavage of HOIP at D348/D387/D390 does not affect its apoptosis-preventing function." This includes the following observations:

- i. Figure 3A and Appendix Figure S2A. In addition to caspases 3 and 6, caspases 8 and 10a are also able to cleave HOIP *in vitro*. By contrast, caspase-7, whilst being active against its substrate PARP-1, is not able to cleave HOIP.
- ii. Figure 3B and Appendix Figure S2B. In MCF7 cells, which do not express caspase-3, HOIP is cleaved upon TRAIL and TNF/CHX stimulation. Therefore, caspase-3 is not required, and hence cannot be the sole responsible caspase, for HOIP cleavage. Whilst knockdown (KD) of caspase-8

nearly abrogated HOIP cleavage, combined KD of caspases 6 and 10 did not affect it. Moreover, KD of caspases 6 and 8 together did not further limit HOIP cleavage as compared to KD of caspase-8 alone. Thus, we conclude that caspase-8 is involved in TRAIL- as well as TNF/CHX-induced HOIP cleavage whereas caspases 6 and 10 are not.

iii. Figure 3C and appendix Figure S2C. KD of caspase-8 in HeLa cells largely reduced HOIP cleavage upon both TRAIL and TNF/CHX stimulation whilst KD of caspase-3 only marginally reduced HOIP cleavage. Thus, caspase-8 is the main caspase responsible for HOIP cleavage upon TRAIL and TNF/CHX stimulation, whilst the role of caspase-3 is minor.

iv. Appendix Figures S2D,E and Figure 1F. In addition, whilst in both K562 (Figure 1F) and HT29 cells (Appendix Figure S2D) caspase-8 activation and HOIP cleavage coincided, activation of effector caspases occurred later. Notably, HOIP is also cleaved in MEFs during TNF-induced apoptosis and this processing is therefore not limited to human cells (Appendix Figure S2E).

v. Figure 3D. We found that HOIP is equally cleaved upon TRAIL stimulation in HCT116 WT and BAX/BAK-DKO cells at early time points even though caspase-3 is not fully activated in BAX/BAK-DKO cells. At later times, however, HOIP seems to be slightly more cleaved in WT compared to BAX/BAK-DKO cells.

In light of these new findings (i to v), we conclude that caspase-8 is the main caspase responsible for HOIP cleavage upon death receptor stimulation and that caspase-3 contributes to it to a lesser extent at later time points.

vi. Figures 3E, F. We have reconstituted K562 HOIP KO cells with HOIP WT and different HOIP mutants. In accordance with the results obtained by Joo *et al.* upon TNF/CHX stimulation (Joo *et al.*, 2016), our analysis reveals that HOIP is also cleaved upon TRAIL stimulation at D348, D387 and D390. Functionally, reconstituting K562 HOIP KO cells with HOIP WT or HOIP D348A/D387A/D390A (HOIP^{AAA}) rescued these cells from TRAIL-induced apoptosis to a similar extent. Therefore, cleavage of HOIP at D348/D387/D390 does not alter its apoptosis-preventing function in TRAIL signalling.

We were pleased that the reviewer shared our interest in determining the impact of HOIP cleavage on gene-activatory signalling. As discussed in the first submission, the cleavage of HOIP could, on the one hand, affect the recruitment of CYLD by removing its PUB domain, and potentially therefore act as a positive regulator of linear ubiquitination in this complex (Draber *et al.*, 2015; Schaeffer *et al.*, 2014; Takiuchi *et al.*, 2014). On the other hand, caspase-dependent cleavage of HOIP would also result in removal of the ZF and NZF1 domains, potentially affecting its interaction with ubiquitin and NEMO (Fujita *et al.*, 2014; Haas *et al.*, 2009; Ikeda *et al.*, 2011). By using reconstituted A549 and HT29 HOIP KO cells, we demonstrate that HOIP cleavage did not affect its ability to promote gene-activatory signalling. These results are shown in the Figures EV4A-D.

Overall, caspase-dependent cleavage of HOIP neither prevents nor enhances its apoptosis-preventing and gene-activatory-promoting function.

In addition to recruiting A20 to receptor complexes, HOIP also interact with CYLD and OTULIN.

However, there is no investigation of the recruitment of CYLD and OTULIN to the TRAIL receptor complexes. This needs to be investigated, in particular because HOIP appears to prevent necroptosis independently of its catalytic activity and CYLD regulates necroptosis after TNF treatment. Additionally, it would be of interest to know if the cleavage of HOIP regulates the association of these deubiquitylases with the signalling complexes.

This is again an interesting point raised by the reviewer. We have compared complex formation in HOIP-proficient and deficient cell lines to address this point. These experiments have led to the following observations:

i. Figures EV2B and C: In A549 and MEFs, we did not detect OTULIN in the different TRAIL signalling complexes studied. By contrast, CYLD is recruited in a HOIP-dependent manner to TRAIL signalling complexes in the presence of zVAD. Whilst we cannot completely exclude the ability of an OTULIN-associated LUBAC pool to be recruited to these complexes, if such an interaction exists, it might be very transient and/or particularly weak, as also discussed by Wagner et al. for TNFR1-SC (Wagner et al, 2016). As discussed in the manuscript, given its usual function towards cell death, we deem highly unlikely that absence of CYLD recruitment in HOIP-deficient cells would account for sensitisation to TRAIL-induced death.

ii. Figures EV2D. Regarding the impact of the cleavage of HOIP on the recruitment of DUBs to the TRAIL signalling complex I, we observed that reconstitution of A549 HOIP KO cells with HOIP WT or the non-cleavable mutant HOIP^{AAA} equally rescued A20 recruitment to the TRAIL complex I. Of note CYLD, which is a direct substrate of caspase-8 and is cleaved upon TRAIL stimulation independently of HOIP expression, is not detected in complex I when caspases are not inhibited. Interestingly, cleavage of HOIP is not required to prevent CYLD recruitment to TRAIL complex I as CYLD was not detected in complex I in HOIP KO A549 cells independently of whether they were reconstituted with HOIP-WT or HOIP^{AAA}. Therefore, caspase-8 prevents CYLD recruitment to TRAIL signalling complexes by directly cleaving it.

It is interesting that HOIP seems to primarily regulate ERK activation after TRAIL treatment, while having very limited effects on other MAP kinases and IκappaB turnover (Figure 6). However, this is only shown in one cell line and needs to be demonstrated also in other cell types used in the study.

We thank the reviewer for this insightful comment suggesting to further define the influence of HOIP on MAP kinase activation. We have now determined the influence of HOIP on ERK, IκBα, but also p38 and JNK in TRAIL and TNF signalling in several cell types:

i. Figures 7B, C, EV3A and figures 7A, EV3B-D, EV4B. We demonstrate that HOIP promotes TRAIL-induced NF-κB activation in five different cell types: HT29, HeLa, A549, K562 and primary Bone-Marrow Derived Macrophages (BMDMs). As included in the first submission, IKK recruitment and activation in complex I is abrogated in the absence of HOIP in both A549 and HeLa cells. Similarly, TNF-induced NF-κB activation is also HOIP-dependent in HeLa and primary BMDMs.

ii. Figure 7C, EV3A and Figures 7A, EV3C, D, EV4B. Regarding the MAPK pathways, upon TNF stimulation HOIP promotes activation of p38, ERK and JNK in HeLa and BMDMs. Upon TRAIL

stimulation, however, p38 and JNK activation is HOIP-dependent only in HT29 cells whereas TRAIL-induced ERK activation is impaired in HOIP-deficient K562, HeLa and HT29 cells.

Thus, HOIP consistently promotes activation of NF- κ B upon TRAIL and TNF-stimulation, and, in certain cell types, participates in MAPK activation.

Additionally, it will be important to determine the functional consequence of decreased ERK vs JNK activation on transcriptional responses/cytokine production. In Figure 6, the role of HOIP on ERK activation was determined in K562 cells while its role in CCL2 production was determined in A549 cells, which prevents any conclusion as to functional importance of ERK activity for CCL2 production.

We appreciate the suggestion to investigate the relative contribution of ERK vs JNK in TRAIL-induced cytokine production. We have evaluated the contribution of MEK1/2, JNK1/2 and, in addition, also p38a/b and IKKa/b to both, TRAIL- and TNF-induced cytokine production in HeLa as well as in A549 cells. We found that inhibition of each of these kinases reduced TRAIL- and TNF-induced cytokine production, with IKKa/b being the kinases required in every case studied. Notably, silencing of p65 reduced both TRAIL- and TNF-induced IL-8 and CCL-2 production. These results are included in the Figures 7F, G and Appendix Figure S2A-C. Regarding the contribution of ERK vs. JNK in cytokine production, we found that inhibition of MEK1/2 is more potent than JNK1/2 inhibition in reducing both TNF- and TRAIL-induced IL-8 production. By contrast, JNK1/2 inhibition is more potent than MEK1/2 inhibition in reducing TRAIL- and TNF-induced CCL2 production. Therefore, depending on the cytokine induced, ERK and JNK activation are differentially required.

We also thank the reviewer for the suggestion to evaluate gene-activatory signalling and cytokine production in the same cell lines. We now include an analysis of both of these aspects in both cell lines, HeLa and A549.

Similar to the result in A549 cells (Figure 7D), HOIP promotes TRAIL-induced cytokine production in HeLa cells (Figure 7E). In HeLa cells, both TRAIL-induced ERK and NF- κ B activation are affected by absence of HOIP (Figure 7A) whereas in A549 cells, solely TRAIL-induced NF- κ B activation is reduced in absence of HOIP (Figure 7B) whilst ERK is neither activated in A549 HOIP KO nor in A549 HOIP control cells upon TRAIL stimulation. Taken together, these results identify NF- κ B activation as required for HOIP-dependent TRAIL-induced cytokine production in A549 and HeLa cells. Moreover, HOIP-dependent ERK activation appears to contribute to TRAIL-induced cytokine production in HeLa cells. This is commented in the results section in the paragraph entitled "HOIP promotes TRAIL-induced gene-activatory signalling and the ensuing cytokine production".

Previously reports indicate that impairment of LUBAC function strongly decreases JNK activation during TNF and CD40L signalling (Haas et al. 2009, Gerlach et al. 2011). Are there differences between TRAIL vs TNF/CD40L that determine this differential requirement for LUBAC in activation of MAPKs? Alternatively, might the observed differences simply be due to cell type differences and not as such differences in TNF and TRAIL signalling? The authors should compare MAP kinase pathways after TRAIL and TNF signalling side-by-side in the same cell lines to resolve this.

To address this point, we have compared the activation of the different kinase pathways mentioned in HOIP-proficient and -deficient HeLa and BMDMs side-by-side upon TRAIL and TNF stimulation (Figures 7A, EV3B-D). Upon TNF stimulation, HOIP promotes JNK, p38, ERK and I κ B α phosphorylation in both HeLa and BMDMs. Upon TRAIL stimulation, I κ B α phosphorylation is HOIP-dependent in both HeLa and BMDMs. In addition, ERK activation is also reduced in TRAIL-stimulated HOIP-deficient HeLa cells whereas the activation of JNK and p38 is neither affected in TRAIL-stimulated HOIP-deficient HeLa cells nor BMDMs. Therefore, activation of the NF- κ B pathway is consistently affected by absence of HOIP regardless of cell type and stimulus. By contrast, the role of HOIP in MAPK activation is both cell type- and stimulus-dependent. Defining the molecular basis for the observed differences between these two different signalling pathways with regards to distinct cell types will require further in-depth investigation and goes beyond the scope of the current study which identifies LUBAC as a major component of TRAIL signalling.

Specific comments:

The claim that the HOIP catalytic activity is "required to prevent TRAIL-induced apoptosis" is not fully supported by the experimental data in Figure 5: In K562 cells, reintroduction HOIP WT partially rescued the cell death phenotype and in MEFs catalytically dead HOIP provided partial protection from TRAIL. Please rephrase the statement to reflect the data.

Figure 5C: Pretreatment with Nec1-s before TRAIL should be included to assess the contribution of necroptosis in MEFs reconstituted with HOIP variants. Without this control it cannot be concluded that HOIP inhibits necroptosis independently of its activity.

We thank the reviewer for pointing this out and have now included the TRAIL/Nec-1s data in Figures 6B and E. Moreover, we have further investigated the modality of TRAIL-induced cell death in the absence of HOIP. Specifically, we now define the requirement of HOIP's catalytic activity in preventing TRAIL-induced **RIPK1 kinase activity-independent apoptosis (point i)**, **necroptosis (point ii)**, and **RIPK1 kinase activity-dependent apoptosis (point iii)**.

i. Amended Figure 6B: K562 HOIP KO cells reconstituted with HOIP WT are as sensitive as control cells to TRAIL-induced cell death (compare to figure 1E). Notably, treatment with the RIPK1 inhibitor Nec-1s does not provide any protection from TRAIL-induced death whilst zVAD fully prevented it. Therefore, HOIP-deficient K562 cells are sensitised to a TRAIL-induced RIPK1-kinase activity-independent apoptosis. This type of cell death cannot be inhibited by re-expressing HOIPC885S. Thus, linear ubiquitination is required for HOIP's function in preventing **RIPK1 kinase activity-independent apoptosis**.

ii. Amended Figure 6E: In HOIP KO MEFs, in the presence of zVAD, TRAIL cannot induce apoptosis but necroptosis. TRAIL-induced necroptosis is equally reduced by re-expressing HOIP WT or HOIP C885S in HOIP KO MEFs. Therefore, linear ubiquitination is not required for HOIP's function in preventing **TRAIL-induced necroptosis**.

iii. As noted by the reviewer, re-expression of HOIP C885S partially rescued HOIP KO MEFs from TRAIL-induced cell death in the absence of any inhibitors. Since Nec-1s prevents TRAIL-induced death of HOIP KO MEFs, these cells are sensitized to TRAIL-induced RIPK1-kinase-dependent

death. In order to define whether HOIP KO MEFs undergo TRAIL-induced necroptosis in the absence of caspase inhibitors, we used the RIPK3 inhibitor GSK'872. Notably, in HOIP KO MEFs, whilst RIPK3 inhibition prevented TRAIL/zVAD-induced death, this inhibitor did not prevent TRAIL-induced cell death in the absence of caspase inhibitor (Figure EV6F). Therefore, HOIP KO MEFs are sensitised exclusively to a RIPK1-kinase-activity-dependent apoptosis and this type of cell death is partially prevented by re-expression of HOIPC885S. Therefore, HOIP limits **RIPK1 kinase-activity-dependent apoptosis** partially depending on its catalytic activity.

Page 1: The acronym "DD" in "TRAIL also triggers DD-dependent..." Should be explained when first used.

We thank the reviewer for noticing this omission and have added the definition of DD accordingly.

Figure 1: Treatment times should be indicated in the figure legend instead of only in the methods section.

We have amended the figure legends accordingly.

Figure 2 and 3A: It is unclear how/if the "DISC-IP" procedure differs from the "Complex I IP" shown in Figures 6 and 7? The DISC-IP procedure is not described in the M&M section.

We thank the reviewer for pointing this out. We are now constantly using the term complex I for designating the TRAIL-R-associated complex throughout the manuscript. Complex I is defined as being a TRAIL-R associated complex, as opposed to complex II, a secondary, TRAIL-R-devoid complex. The current dogma stipulates that complex I exclusively triggers death and has therefore been referred to as the "Death-Inducing Signalling Complex (DISC)" since its initial discovery. The procedure for the isolation of the DISC and complex I is thus exactly the same as they are both terms designating the same complex. On the basis of the findings presented in this manuscript, we suggest that the TRAIL-R-associated complex, which also contains gene-activatory signalling molecules, should be referred to as the complex I of TRAIL signalling, as the term "DISC" only represents part of its function.

On Page 7 the authors refer to their article (Draber et al. 2015) showing that A20 is recruited to the TNFR1-SC via linear ubiquitin. The original papers showing that A20 is recruited to receptor complexes via linear ubiquitin should also be acknowledged: Verhelst et al. 2012 EMBOJ, Tokunaga et al. 2012 EMBOJ.

We apologise for this omission and thank the reviewer for pointing this out. We now refer to these studies at this point of the manuscript.

Referee #2:

The relevance of the LUBAC and its activity has been intensively studied in recent years in context of TNFR1. However, there is only one study (without in depth analysis) so far (Zhang et al., Cell Signal. 2015 Feb;27(2):306-14) which addressed the role of the HOIP/LUBAC in signaling by the

TNFR1-related TRAIL death receptors. The manuscript by Lafont fills this gap and demonstrates i) that the LUBAC is recruited to the plasma membrane associated TRAILR2 complex and is also part of the secondarily formed TRAIL induced cytoplasmic complex II, ii) that the LUBAC inhibits cell death induction by TRAIL but promotes TRAIL-induced NF κ B/ERK signaling and iii) that LUBAC recruitment to the TRAILR2 signaling complex occurs downstream of FADD, caspase-8 and cIAPs and is required for bridging the TRAILR2 signaling complex with the IKK complex.

TNFR1 and the TRAIL death receptors share a death domain but based on the published data available so far, they differ considerably in the signaling complexes that are formed on the DDs after receptor stimulation. TNFR1 efficiently recruits TRADD, RIP, TRAF2, the LUBAC and the IKK complex but shows no recruitment of FADD and caspase-8. In contrast, TRAILR2 efficiently recruits FADD and caspase-8 but shows only a poor recruitment of TRAF2 and RIP. The study of Lafont et al, now modifies this picture and not only demonstrates significant recruitment of TRAF2 and RIP1 to TRAILR2 but also that this enables co-recruitment of the LUBAC and the IKK complex. The data are therefore not only novel but also of broad and general interest for the death receptor field.

I have only a few minor concerns/comments on the manuscript:

We thank the reviewer for acknowledging the significance and broad impact of our study and for sharing our excitement about these novel findings. We particularly thank this reviewer for acknowledging the differences between TRAIL and TNF signalling and how our study helps broaden our understanding of the sequence of events in TRAIL signalling and how they differ from those during TNF signalling. We also thank the reviewer for raising relevant points which we have now addressed as explained in the following paragraphs.

Figure 2A - The receptor bands are labeled with TRAILR1. The appearance of the bands and the text suggest, however, that TRAILR2 is shown.

The staining shown is TRAIL-R1. We are also puzzled by the second band observed. However, we reproducibly see these two bands when we stain for TRAIL-R1 in HeLa cells in lysates, but also in complex I.

Paragraph "LUBAC is recruited" - The second part of the first sentence "As HOIP prevents TRAIL-induced cell deathas this is the receptor-associated protein complex we and others previously showed initiates this signaling outcome. " is incomprehensible. I suggest rephrasing or deletion.

We thank the reviewer for pointing this out and agree that the initial version was too convoluted. We have corrected the sentence which now reads as follows: "We and others previously identified the TRAIL-R1/2-associated complex I as a cell death-initiating platform (Kischkel et al, 2000; Sprick et al, 2000). As HOIP prevents TRAIL-induced cell death, we hypothesised that LUBAC might form part of this TRAIL-R-associated DISC."

Figure 3B - The FADD IP of non-stimulated cells shows significant amounts of RIP and RIP3. It should be clarified whether this is due to constitutive FADD-RIP-RIP3 complex formation or whether this represents non-specific binding to the protein G beads.

We agree with the reviewer regarding the necessity to provide this type of control. We have now repeated this experiment adding a beads-only control which shows that the unspecific binding observed in the FADD-IP samples from unstimulated cells is due to unspecific binding of these proteins to the beads. This is now included in the Figure EV2C.

TRAILR2 and complex II IPs performed with non-stimulated cells frequently contain TRAF2 and RIP. I assume that this represents non-specific binding to the protein G beads and not specific association. This should be resolved for the reader by showing control IPs with protein G beads only versus TRAILR2 and complex II IPs.

We thank the reviewer for suggesting the addition of these controls. We have now included a Figure EV2B. This indeed shows, as anticipated by the reviewer, that the background observed in complex I and II in unstimulated samples for RIPK1 and TRAF2, is largely due to unspecific binding of these proteins to the beads.

Figure EV1 F - A549 cells should be included here as this cell type is used in most experiments in the manuscript.

We have now repeated this experiment, also including A549. As shown, HeLa, K562 and A549 do not express RIPK3 (former EV1F, Appendix Figure S1F) and are thereby unable to undergo necroptosis.

A scheme of TRAILR2 signaling summarizing the findings of the manuscript could be helpful for the reader.

We thank the reviewer for this very relevant suggestion. A model is now provided in the Figure EV8C.

Figure 6D and 7 - There is significant processing of FLIP-L and caspase-8 in the IPs although TRAIL + the caspase inhibitor ZVAD have been used for stimulation. This should be commented/explained. As DISC-associated caspase activity is obviously not fully blocked, the authors cannot conclude in the discussion that caspase-8 activity has no relevance for LUBAC recruitment etc.

We have consistently seen that zVAD cannot efficiently prevent the first cleavage of caspase-8 in TRAIL complex I and II, as previously observed by others within TRAIL complex I (Harper et al, 2001), or in CD95 complex I (Geserick et al, 2009). We therefore completely agree with the reviewer that this caspase still does display a DISC-restricted partial activation in the presence of zVAD and have removed the corresponding sentence in the discussion.

It should be indicated throughout the manuscript whether FLIP-L or FLIP-S or both are meant when the term FLIP is used.

This is a valid point. We have now amended the text and specified in each case which isoforms of FLIP we refer to.

Discussion last paragraph - "As a consequence of TRAIL-induced gene activation several cytokines are produced (Tang et al. 2009; Trauzold et al., 2006; Varfolomeev et al., 2005)." I recommend to cite here earlier papers showing this already many years before (e.g. Harper et al., J Biol Chem. 2001 Sep 14;276(37):34743-52 or Wajant et al., J Biol Chem. 2000 Aug 11;275(32):24357-66).

The reviewer is completely correct and we sincerely apologize for this omission. This paragraph in the discussion has now been substantially rewritten and we have added the references mentioned by the reviewer in the results section relative to TRAIL-induced gene activation and cytokine production.

The authors should discuss Zhang et al., (Cell Signal. 2015 Feb;27(2):306-14) which analyzed TRAIL-induced NFkappaB signaling in HOIP deficient A20 cells and observed a HOIP-dependent and a HOIP-independent mode of NFkappaB signaling!

We agree with the reviewer that this article has to be referred to in our manuscript and thank them for pointing this out to us. We have now included this reference in the discussion.

References:

Draber P, Kupka S, Reichert M, Draberova H, Lafont E, de Miguel D, Spilgies L, Surinova S, Taraborrelli L, Hartwig T, Rieser E, Martino L, Rittinger K, Walczak H (2015) LUBAC-Recruited CYLD and A20 Regulate Gene Activation and Cell Death by Exerting Opposing Effects on Linear Ubiquitin in Signaling Complexes. *Cell Rep* **13**: 2258-2272

Fujita H, Rahighi S, Akita M, Kato R, Sasaki Y, Wakatsuki S, Iwai K (2014) Mechanism underlying IkkappaB kinase activation mediated by the linear ubiquitin chain assembly complex. *Mol Cell Biol* **34**: 1322-1335

Geserick P, Hupe M, Moulin M, Wong WW-L, Feoktistova M, Kellert B, Gollnick H, Silke J, Leverkus M (2009) Cellular IAPs inhibit a cryptic CD95-induced cell death by limiting RIP1 kinase recruitment. *J Cell Biol* **187**: 1037-1054

Haas TL, Emmerich CH, Gerlach B, Schmukle AC, Cordier SM, Rieser E, Feltham R, Vince J, Warnken U, Wenger T, Koschny R, Komander D, Silke J, Walczak H (2009) Recruitment of the linear ubiquitin chain assembly complex stabilizes the TNF-R1 signaling complex and is required for TNF-mediated gene induction. *Mol Cell* **36**: 831-844

Harper N, Farrow SN, Kaptein A, Cohen GM, MacFarlane M (2001) Modulation of tumor necrosis factor apoptosis-inducing ligand-induced NF-kappa B activation by inhibition of apical caspases. *J Biol Chem* **276**: 34743-34752

Ikeda F, Deribe YL, Skanland SS, Stieglitz B, Grabbe C, Franz-Wachtel M, van Wijk SJ, Goswami P, Nagy V, Terzic J, Tokunaga F, Androulidaki A, Nakagawa T, Pasparakis M, Iwai K, Sundberg JP, Schaefer L, Rittinger K, Macek B, Dikic I (2011) SHARPIN forms a linear ubiquitin ligase complex regulating NF-kappaB activity and apoptosis. *Nature* **471**: 637-641

Joo D, Tang Y, Blonska M, Jin J, Zhao X, Lin X (2016) Regulation of Linear Ubiquitin Chain Assembly Complex by Caspase-Mediated Cleavage of RNF31. *Mol Cell Biol* **36**: 3010-3018

Kischkel FC, Lawrence DA, Chuntharapai A, Schow P, Kim KJ, Ashkenazi A (2000) Apo2L/TRAIL-dependent recruitment of endogenous FADD and caspase-8 to death receptors 4 and 5. *Immunity* **12**: 611-620

Schaeffer V, Akutsu M, Olma MH, Gomes LC, Kawasaki M, Dikic I (2014) Binding of OTULIN to the PUB domain of HOIP controls NF-kappaB signaling. *Mol Cell* **54**: 349-361

Sprick MR, Weigand MA, Rieser E, Rauch CT, Juo P, Blenis J, Krammer PH, Walczak H (2000) FADD/MORT1 and caspase-8 are recruited to TRAIL receptors 1 and 2 and are essential for apoptosis mediated by TRAIL receptor 2. *Immunity* **12**: 599-609

Takiuchi T, Nakagawa T, Tamiya H, Fujita H, Sasaki Y, Saeki Y, Takeda H, Sawasaki T, Buchberger A, Kimura T, Iwai K (2014) Suppression of LUBAC-mediated linear ubiquitination by a specific interaction between LUBAC and the deubiquitinases CYLD and OTULIN. *Genes Cells* **19**: 254-272

Wagner SA, Satpathy S, Beli P, Choudhary C (2016) SPATA2 links CYLD to the TNF-alpha receptor signaling complex and modulates the receptor signaling outcomes. *EMBO J* **15**

2nd Editorial Decision

13 February 2017

Thank you for submitting your revised manuscript for our consideration. Referees 1 and 2 have now assessed it once more, and I am happy to inform you that both consider the study substantially improved and all original concerns satisfactorily addressed. We have therefore now accepted it for publication in The EMBO Journal.

Corresponding Author Name: Henning Walczak

Manuscript Number: EMBOJ-2016-95699